# Chemical labelling for visualizing native AMPA receptors in live neurons

Sho Wakayama[1],[*], Shigeki Kiyonaka[1],[*], Itaru Arai[2], Wataru Kakegawa[2], Shinji Matsuda[2,3,4], Keiji Ibata[2], Yuri L. Nemoto[5], Akihiro Kusumi[5,6], Michisuke Yuzaki[2,7] & Itaru Hamachi[1,7]

The location and number of neurotransmitter receptors are dynamically regulated at postsynaptic sites. However, currently available methods for visualizing receptor trafficking require the introduction of genetically engineered receptors into neurons, which can disrupt the normal functioning and processing of the original receptor. Here we report a powerful method for visualizing native α-amino-3-hydroxy-5-methyl-4-isoxazolepropionic acid (AMPA)-type glutamate receptors (AMPARs) which are essential for cognitive functions without any genetic manipulation. This is based on a covalent chemical labelling strategy driven by selective ligand-protein recognition to tether small fluorophores to AMPARs using chemical AMPAR modification (CAM) reagents. The high penetrability of CAM reagents enables visualization of native AMPARs deep in brain tissues without affecting receptor function. Moreover, CAM reagents are used to characterize the diffusion dynamics of endogenous AMPARs in both cultured neurons and hippocampal slices. This method will help clarify the involvement of AMPAR trafficking in various neuropsychiatric and neurodevelopmental disorders.

[1] Department of Synthetic Chemistry and Biological Chemistry, Graduate School of Engineering, Kyoto University, Nishikyo-ku 615-8510, Japan. [2] Department of Physiology, School of Medicine, Keio University, Tokyo 160-8582, Japan. [3] Department of Engineering Science, Graduate School of Informatics and Engineering, University of Electro-Communication, Tokyo 182-8585, Japan. [4] PRESTO, Japan Science and Technology Agency, Saitama 332-0012, Japan. [5] Institute for Frontier Medical Sciences, Kyoto University, Kyoto 606-8507, Japan. [6] Institute for Integrated Cell-Material Sciences (WPI-iCeMS), Kyoto University, Kyoto 606-8507, Japan. [7] CREST(Core Research for Evolutional Science and Technology, JST), Chiyodaku, Tokyo 102-0075, Japan. * These authors contributed equally to this work. Correspondence and requests for materials should be addressed to S.K. (email: kiyonaka@sbchem.kyoto-u.ac.jp) or to I.H. (email: ihamachi@sbchem.kyoto-u.ac.jp).

Various neurotransmitter receptors located on postsynaptic membranes regulate important brain functions, including memory, learning, language and reasoning. The location and expression levels of each receptor are not only determined precisely during development but also dynamically regulated throughout adulthood by various receptor trafficking mechanisms such as endocytosis, exocytosis and lateral diffusion in response to changes in neuronal activity. Therefore, to understand whether and how neurotransmitter receptor functions change under both physiological and pathological conditions, it is essential to visualize and characterize how receptor trafficking is regulated or dysregulated in living physiological neuronal circuits.

One way to perform live imaging of receptor trafficking is to express in neurons a genetically engineered receptor fused with fluorescent proteins, such as green fluorescent protein (GFP)[1]. A drawback of this method is that receptors located on the cell surface cannot be distinguished from those in intracellular pools. Because intracellular pools are acidic, the use of a pH-sensitive variant of GFP (pHluorin or super-ecliptic pHluorin (SEP)) that emits minimal fluorescence in acidic environments enables selective visualization of cell surface receptors[2–4]. However, the relatively large size of the fused fluorescent proteins (25 kDa) can disturb the membrane trafficking and ion channel activity of the receptor[5,6].

An alternative approach is to chemically label cell-surface receptors. For this purpose, genetically engineered receptors fused with protein tags such as SNAP and Halo are expressed in neurons. Receptors located on the cell surface are then enzymatically and covalently labelled with fluorescent probes such as fluorescein and Alexa dyes[7,8]. To reduce the size of the protein tags (20–33 kDa), a complementary recognition pairs comprising a short peptide tag (1–3 kDa) and a small molecular probe are also being developed[9–13]. A combination of bio-orthogonal chemistry and genetic incorporation of a non-naturally occurring amino acid is also claimed to effectively label cell-surface receptors with minimal structural disturbance[14]. Although these chemical labelling approaches are powerful, there remains a concern that the introduction of any non-native or genetically modified receptors to neurons could disturb natural receptor trafficking, in which the number and localization of endogenous receptors are precisely regulated.

Ideally, endogenous receptors would be visualized without any recourse to genetic manipulation. Antibodies that recognize the extracellular domain of receptors can be used to label cell-surface receptors in extrasynaptic sites in dissociated neurons but often are unable to reach synaptic sites, which are densely populated with an array of synaptic proteins[15,16], especially in native brain tissues. Instead, small fluorophore-conjugated ligands, which selectively bind to the target receptors, have been developed to visualize endogenous receptors[17]. However, this approach is limited by the reversible non-covalent interaction of ligands with target receptors and its antagonistic action on receptor function. To overcome these problems, improved affinity-based covalent labelling methods including ligand-directed chemistry for endogenous proteins have been developed recently[18–22]. However, there have been few demonstrations of this technology for endogenous neurotransmitter receptors, and those that do exist involve complicated experimental procedures[22]. Furthermore, no studies have reported successful live imaging of endogenous neurotransmitter receptors in brain tissues.

Fast excitatory neurotransmission in the vertebrate central nervous system is achieved mainly via the α-amino-3-hydroxy-5-methyl-4-isoxazolepropionic acid (AMPA)-activated subtype of the glutamate receptor family (AMPAR). Long-lasting changes in the number of postsynaptic AMPARs[23,24] are considered as the basis of learning and memory. Nevertheless, conflicting findings for AMPAR trafficking have been reported[25–33], most likely reflecting the limitations of currently available methods. For example, overexpression of a single AMPAR subtype[31,32] and pH changes in intracellular pools[3,53] could have affected the differences.

In the present study, we report a promising traceless protein labelling method that effectively tethers various small fluorescent probes to endogenous AMPARs located at the cell surface without affecting AMPAR function. This method will be a powerful and useful tool to visualize and precisely evaluate the dynamics of endogenous AMPARs in not only cultured dissociated neurons but also in brain slices.

## Results

**General strategy for chemical labelling of AMPARs**. For covalent attachment of small chemical probes to AMPARs, we applied ligand-directed acyl imidazole (LDAI) chemistry, a traceless protein labelling method[19,20]. LDAI-based chemical labelling is driven by selective ligand-protein recognition, which facilitates an acyl substitution reaction of labelling reagents to nucleophilic amino acid residues (Lys, Ser or Tyr) located near the ligand-binding domain (Fig. 1a). Here we carefully designed labelling reagents for AMPARs by taking into consideration the selectivity of the affinity ligand, the orientation of the acyl imidazole group, and the total charges of the labelling reagents. We chose 6-pyrrolyl-7-trifluoromethyl-quinoxaline-2,3-dione (PFQX) as a ligand, because PFQX exhibits a sufficient affinity ($K_i$ value of 170 nM) and selectivity for AMPARs over other glutamate receptors, including N-methyl-D-aspartate (NMDA) and kainate receptors[34,35] (Fig. 1b). In addition, this negatively charged ligand is relatively hydrophilic, which offers the possibility of selective labelling of cell-surface AMPARs by suppressing permeation of the labelling reagents into live neurons. The pyrrolyl moiety of PFQX was assumed to be accessible to the surface of the ligand-binding domain based on X-ray structural analysis of an AMPAR bound with ZK200775, an antagonist similar to PFQX[36]. Thus, a variety of probes were connected to this pyrrolyl moiety of PFQX via a reactive acyl imidazole linker. We prepared labelling reagents with various spacers between the reactive acyl imidazole unit and the ligand to finely control the position of the acyl imidazole unit on the AMPAR surface, and we termed this series of probes 'chemical AMPAR modification' (CAM) reagents (Fig. 1b). Notably, the labelling procedure using these reagents is very simple, involving only brief incubation and washing procedures. In addition, the excess labelling reagents and the cleaved ligand moiety can be readily washed out (Fig. 1a).

**Chemical labelling of AMPARs in HEK293T cells**. Covalent labelling of AMPARs was initially examined in live HEK293T cells transiently transfected with the GluA2 (GluA2(R)) subtype, a major subtype of AMPARs in brains. The labelling was performed by incubating cells with one of three CAM reagents (**CAM1(OG)**, **CAM2(OG)**, and **CAM3(OG)**) at 17 °C to suppress the internalization of labelled AMPARs[37]. Western blot analyses of the cell lysate showed a single strong band corresponding to the AMPAR (110 kDa) in cells expressing GluA2 but not in cells transfected with a control vector (lanes 3 and 5 in Fig. 2a). This band was absent in the presence of NBQX, a competitive antagonist of AMPARs, indicating that labelling was facilitated by an affinity-driven proximity effect (lanes 3 and 4 in Fig. 2a). The AMPAR labelling efficiency was dependent on the spacer length of the reagents, with **CAM2(OG)** the most effective (lanes 1, 3, 6 in Fig. 2a). The labelling sites on the AMPARs were determined

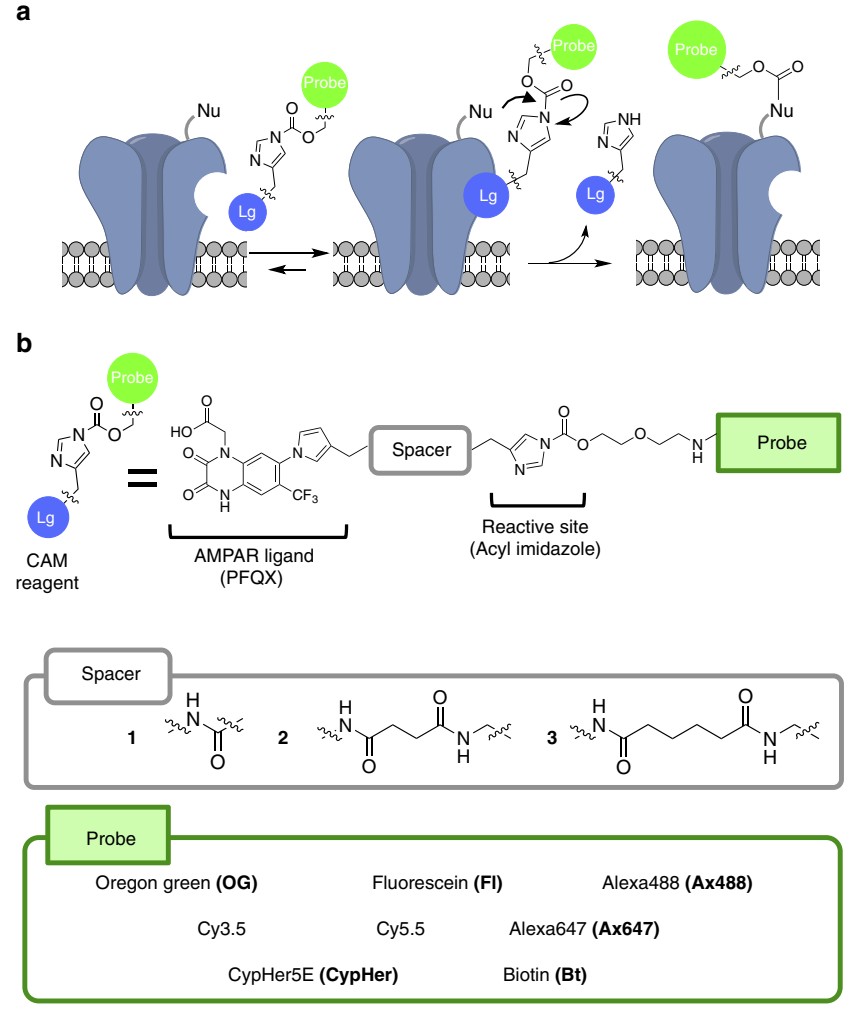

**Figure 1 | Chemical labelling of AMPARs using CAM reagents.** (**a**) Schematic illustration of chemical labelling of AMPARs driven by the selective ligand-protein recognition using CAM reagents. Lg, ligand moiety; Nu, nucleophilic amino acid residue. (**b**) Chemical structures of the CAM reagents. The detailed chemical structures are shown in Supplementary Methods.

using a recombinant version of the ligand-binding domain of GluA2 (ref. 38), which revealed that Lys470 located at the entrance of the ligand-binding pocket was the primary site labelled with **OG** (Fig. 2b and Supplementary Fig. 1).

Live-cell confocal imaging of GluA2-expressing HEK293T cells labelled with **CAM2(OG)** clearly showed that the fluorescence was observed predominantly at the plasma membrane (Fig. 2c), although immunostaining after fixation and permeabilization showed that GluA2 was mainly localized in intracellular compartments (Supplementary Fig. 3). As expected, **CAM2(OG)** did not permeate into live cells (Supplementary Fig. 4a), and selectively labelled cell-surface GluA2 (Supplementary Fig. 4b). In good agreement with the western blot analysis, the fluorescence was not observed in the presence of NBQX or in vector-transfected cells (Fig. 2c). Interestingly, whereas several bands were observable on the western blots of GluA2-transfected cells using an anti-GluA2/3 antibody, only a single labelled band was apparent when using the anti-Fl/OG antibody (Fig. 2a). Reprobing the blots showed that the labelled protein could be assigned to the highest molecular weight (100–110 kDa) among the GluA2-positive bands (Supplementary Fig. 5). These results indicated that cell-surface GluA2, properly modified by post-translational processes, was selectively labelled by the cell-impermeable **CAM2(OG)** in HEK293T cells.

We also examined whether AMPARs could be labelled with **CAM2** reagents bearing different chemical probes (Fig. 1b). We found that GluA2 in HEK293T cells was successfully and selectively modified with fluorescent probes exhibiting different fluorescent wavelengths or properties (fluorescein, Alexa488, Cy3.5, Cy5.5 and Alexa647), including a pH-sensitive probe (CypHer5E) (Supplementary Fig. 6). In addition, biotin was also selectively attached to AMPARs using **CAM2(Bt)**, which was subsequently visualized by the streptavidin-fluorophore conjugate (SAv-Ax555). These results indicate the broad applicability of **CAM2** as a tool to study various aspects of AMPAR trafficking.

A functional AMPAR is a tetramer consisting of a combination of four subunits, GluA1–GluA4 (refs 23,24,39). To examine selectivity of **CAM2** reagents to GluA subunits, HEK293T cells were transfected with each subunit (GluA1, GluA2, GluA3 or GluA4). WB analysis revealed that **CAM2** successfully labelled GluA2, GluA3 and GluA4 but not GluA1 (Supplementary Fig. 7a–d), although GluA1 was prominently expressed on the cell surface like GluA2 (Supplementary Fig. 7e). The phylogenetic tree indicates the low homology of GluA1 among the AMPAR subunits (GluA1–4; Supplementary Fig. 7f), and two of the three labelling sites identified for GluA2 are not conserved in GluA1 (Fig. 2b and Supplementary Fig. 7g). Such difference in the microenvironment of the entrance of the ligand-binding pocket may inhibit the GluA1 labelling by **CAM2**.

One of the advantages of our ligand-directed chemistry over conventional affinity-based techniques is that the ligand moiety can be removed by a simple washing procedure after chemical labelling. This 'traceless' nature is expected to allow receptors to recover their original functions after labelling. Nevertheless, attachment of chemical probes near the entrance of the ligand-binding pocket could potentially affect the function of AMPARs. To address this concern, we first performed $Ca^{2+}$ imaging and found that **CAM2(OG)** did not affect $Ca^{2+}$ responses in HEK293T cells expressing $Ca^{2+}$-permeable AMPARs (GluA2(Q))[40–42] (Supplementary Fig. 8). Next, the influence of the chemical labelling on channel kinetics was evaluated with a fast glutamate application technique using a piezo element.

When 1 mM glutamate was applied to outside-out patches excised from HEK293 cells expressing GluA2(Q), transient inward currents with rapid channel activation and desensitization kinetics characteristic for AMPARs were observed (Fig. 2d). Incubation of HEK293 cells with 10 μM **CAM2** reagents for 4 h at 17 °C, under which conditions almost all cell-surface AMPARs were labelled with the fluorophore (Supplementary Fig. 2), did not affect the desensitization (Fig. 2d,e) or activation (Fig. 2d,f) kinetics of AMPAR-mediated currents. Furthermore, the dose-response relationship of the glutamate-induced currents was unchanged after labelling with the fluorophore (Fig. 2g). Together, these data indicate that GluA2, GluA3 and GluA4 subunits of AMPARs on the plasma

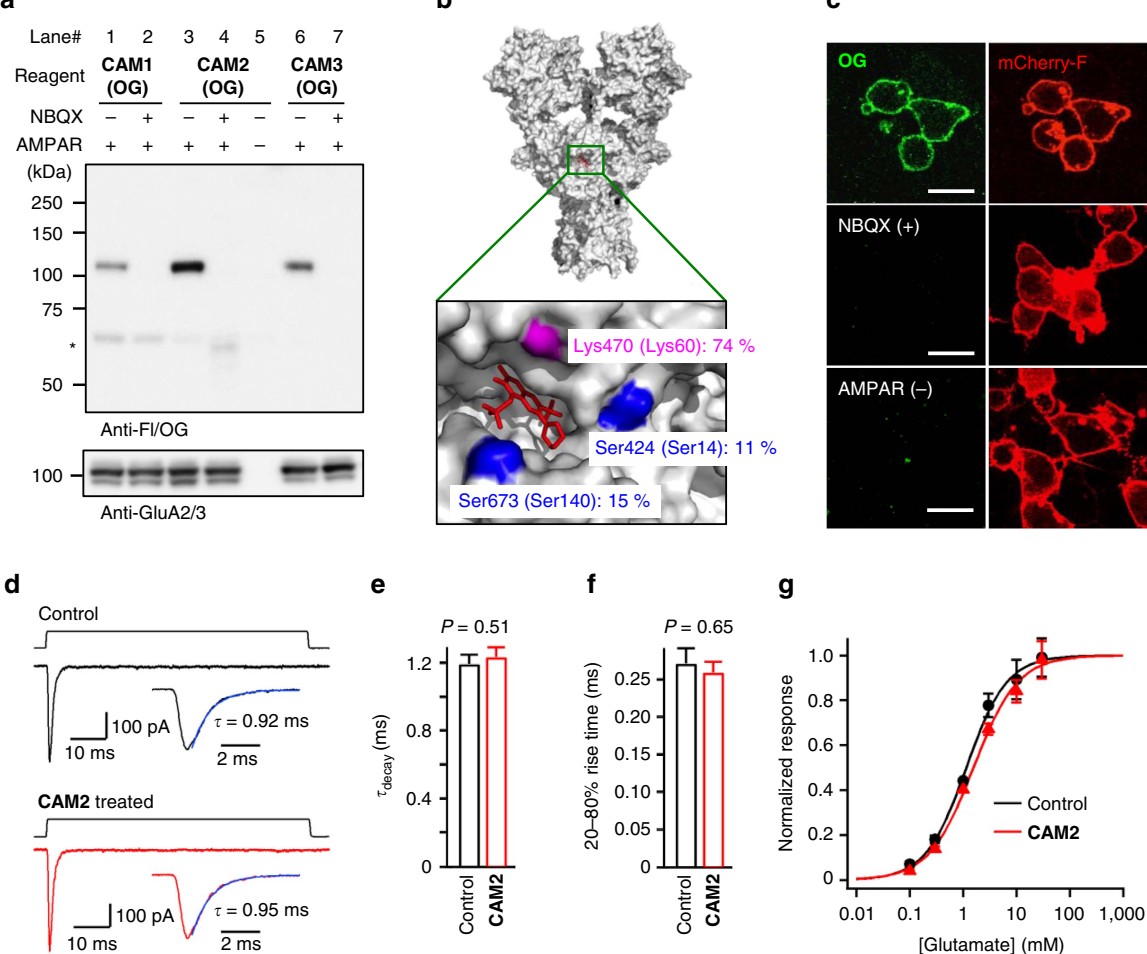

**Figure 2 | Chemical labelling of AMPARs recombinantly expressed in HEK293T cells. (a**) Western blot analyses of HEK293T cells labelled by CAM reagents. HEK293T cells transfected with GluA2[flop](R) (AMPAR( + )) or vector control (AMPAR( − )) were treated with 2 μM **CAM1(OG)**, **CAM2(OG)**, or **CAM3(OG)** in the presence or absence of 50 μM NBQX in serum free DMEM. The cell lysates were analyzed by western blot analysis using anti-Fl/OG or anti-GluA2/3 antibody. * indicates non-specific labelling to bovine serum albumin included in culture medium. (**b**) Identification of labelling site of **CAM2(OG)** to AMPARs using the recombinant ligand-binding domain of GluA2 (GluA2(S1S2J)). Labelled amino acid residue and the labelling yield are indicated in the crystal structure of full-length GluA2 tetramer (PDB: 3KG2). ZK200775, a competitive antagonist is shown as red stick. Residue numbers of full-length GluA2 (NP_058957) are shown. Residue numbers in GluA2(S1S2J) are also shown in parenthesis. For details, see Supplementary Fig. 1. (**c**) Confocal live imaging of HEK293T cells labelled with **CAM2(OG)**. Labelling was conducted as described in **a**. Labelling yield to surface-exposed GluA2 in this condition was determined as 62.0 ± 2.4% (Supplementary Fig. 2). mCherry-F was utilized as a transfection marker. Scale bars, 20 μm. (**d–g**) Effects of chemical labelling on AMPAR function. Current responses to 80-ms applications of glutamate using piezo element (during upper step pulses) were obtained in outside-out patches from HEK293 cells expressing GluA2[flop](Q) with or without the chemical labelling (For details, see Methods section). In **d**, Representative current traces to 1 mM glutamate are shown. In **e,f**, effects of chemical labelling on desensitization ($\tau_{decay}$) or activation kinetics (20–80% rise time) by 1 mM glutamate are shown (n = 13–17). In **g**, dose-response relationships are shown. Peak amplitudes at various glutamate concentrations were fit to the logistical function $a/(1 + (EC_{50}/[glutamate])^{nH})$, where $a$ is maximal amplitude, $EC_{50}$ is the concentration causing a 50% maximum response and nH is Hill coefficient. n = 7–9. The responses were normalized by a Wilcoxon rank test indicates that significant differences were not observed in any points with or without the chemical labelling. Data are represented as mean ± s.e.m.

membrane can be selectively labelled near the ligand-binding domain with a small fluorophore using CAM reagents under live-cell conditions with negligible disturbance of receptor function.

**Visualization of endogenous AMPARs in cultured neurons**. We next examined whether **CAM2** can be applicable to native AMPARs in cultured neurons. Cultured cortical neurons were incubated with **CAM2(OG)** at 17 °C, and the **OG** labelling was evaluated using western blot methods. As shown in Fig. 3a, a single protein band corresponding to that of the AMPAR (100–110 kDa) was clearly detected in the presence of **CAM2(OG)**, and this band was absent in the presence of the competitive inhibitor, NBQX. Importantly, this labelling was not blocked in the presence of an NMDA receptor inhibitor (AP5) or a kainite receptor inhibitor ([2S,4R]-4-methyl glutamate (4MG)), implying that the labelling was selective to AMPARs among the ionotropic glutamate receptor family (Fig. 3b). In addition, immunoprecipitation using anti-Fl/OG antibody showed that covalent attachment of the fluorophore was selective for the AMPARs, with none observed for NMDA or kainite receptors (Fig. 3c). Similarly, native neuronal AMPARs were selectively labelled with other **CAM2** reagents bearing different probes (**CAM2(Fl)**, **CAM2(Ax488)** and **CAM2(Bt)**) (Fig. 3a). Although a lack of probe-specific antibodies precluded western blot analysis for the other probes listed in Fig. 1b, these results indicate that **CAM2** could specifically label native AMPARs in cultured neurons.

Confocal microscopic live imaging of cultured hippocampal neurons labelled with **CAM2(Fl)** showed punctate **Fl** signals along the dendrites (Supplementary Figs 9a and 10). Such punctate **Fl** signals were absent in neurons treated with **PFQX-Fl**, an analog of **CAM2(Fl)** lacking the reactive acyl imidazole moiety for covalent labelling (Supplementary Fig. 9b,c), suggesting that these signals correspond to AMPARs covalently labelled by **CAM2(Fl)**. To examine whether the **Fl** signals were derived from the cell surface AMPARs, fluorescence quenching experiments were conducted using the vital dye trypan blue[43–45]. Similar to the results of HEK293T cells in which trypan blue selectively quenched the fluorescence from the surface-exposed AMPARs but not the internalized ones (Supplementary Fig. 11a–c), addition of trypan blue largely abolished the fluorescence immediately after **CAM2** labelling in cultured hippocampal neurons (Supplementary Fig. 11d–g). These results indicate that the surface-exposed AMPARs were predominantly labelled by our methods. To characterize the **Fl** signals in details, we next performed conventional immunohistochemical analyses on hippocampal neurons labelled by **CAM2(Fl)** after fixation and permeabilization. Confocal microscopy images showed punctate **Fl** signals located on protrusions along dendrites immunopositive for microtubule-associated protein 2 (MAP2; Fig. 3d,f). The **Fl** signals merged well with punctate immunopositive signals for postsynaptic density protein 95 (PSD95; Fig. 3h), and also broadly colocalized with immunoreactivity for GluA2/3 (Fig. 3e,g). Importantly, immunostained punctate signals of surface AMPARs co-localized well with the **Fl** signals (Supplementary Fig. 12). These results indicate that the punctate **Fl** signals likely correspond to chemically labelled synaptic AMPARs located on dendritic spines, and that the vast majority of GluA2/3 immunoreactivity in dendritic shafts reflects AMPARs in intracellular compartments. Besides, similar fluorescent images were obtained in hippocampal neurons treated with **CAM2(Ax488)** (Supplementary Fig. 13), indicating that endogenous AMPARs can be visualized with different kinds of fluorophores using CAM reagents at excitatory synapses in cultured neurons.

We next sought to follow the trafficking of **Fl**-labelled AMPARs during long-term depression (LTD), a well-known synaptic plasticity[23,24]. To apply chemically induced form of LTD (chemLTD)[46], the labelled hippocampal neurons were exposed with NMDA in a short period, and fluorescent changes of **Fl**-labelled AMPARs were visualized by confocal live imaging. As shown in Supplementary Fig. 14, the fluorescence decrease was observed in punctate regions after brief application of NMDA. Taking into consideration of the pH sensitivity of **Fl**-labelled AMPARs on cell surface (Supplementary Fig. 15), the fluorescent change implies the internalization of AMPARs into acidic intracellular endosomes after chemLTD, which is in good agreement with previous reports[3].

**Visualization of endogenous AMPARs in brain tissues**. We subsequently examined whether **CAM2** can successfully label AMPARs in their native three-dimensional environment in hippocampal and cerebellar tissue slices, which include a large number of glial cells (astrocytes, oligodendrocytes and microglial cells). Freshly prepared hippocampal slices (acute slices) were labelled with **CAM2(Fl)** or **CAM2(Ax488)** and evaluated by western blot analysis. As shown in Fig. 4a, a single strong band corresponding to AMPARs was observed in the presence of these labelling reagents, and this band disappeared by co-incubation with NBQX, indicating specific labelling of AMPARs in brain tissues.

We next performed imaging of endogenous AMPARs in brain slices using **CAM2** reagents. To visualize neuronal profiles, mCherry was expressed in P18–22 mice using a neuron-specific lentiviral vector. Acute hippocampal slices were prepared and labelled by **CAM2(Fl)** and fluorescent images were obtained under live conditions. As shown in Fig. 4b, punctate **Fl** signals were observed along mCherry-positive dendrites, and the **Fl** signals were not observed when slices were co-incubated with **CAM2(Fl)** and NBQX (Supplementary Fig. 16a) or were treated with **PFQX-Fl** (Supplementary Fig. 16b). These results further demonstrate specific covalent labelling of AMPARs by **CAM2(Fl)**. Conventional immunohistochemical analysis of hippocampal slices labelled by **CAM2(Fl)** after fixation and permeabilization revealed that punctate **Fl** signals were located on protrusions along dendrites immunoreactive for MAP2 (Supplementary Figs 17 and 18) and were co-localized with GluA2/3 immunoreactivity (Fig. 4c). Similarly, acute cerebellar slices incubated with **CAM2(Fl)** showed moderate and strong **Fl** signals in the granular and molecular layers, respectively; these signals were also largely co-localized with GluA2/3 immunoreactivity (Supplementary Fig. 19). Together, these results demonstrate successful **CAM2(Fl)** labelling of native synaptic AMPARs.

To evaluate the permeability of **CAM2** reagents to brain tissues, we performed Z-axis scanning of labelled slices. Confocal and two-photon microscopic analyses revealed that the fluorescent intensities of the punctate **Fl** signals were detectable throughout the slices with no significant decay in signal intensity (Fig. 4d,e). In contrast, GluA2/3 immunoreactivity was observed only at the surface of the slices, with negligible signal from the deeper regions due to the low tissue penetration of the antibodies (150–200 kDa), even under permeabilized conditions (with 0.2% Triton). These results highlight a major advantage of **CAM2** reagents: their remarkable ability to penetrate into brain tissues.

**Properties of CAM2-labelled native AMPARs in brain tissues**. To address a concern that synaptic AMPAR function may be modified by incubation with **CAM2** reagents, we next performed whole-cell patch-clamp recordings from Purkinje cells in

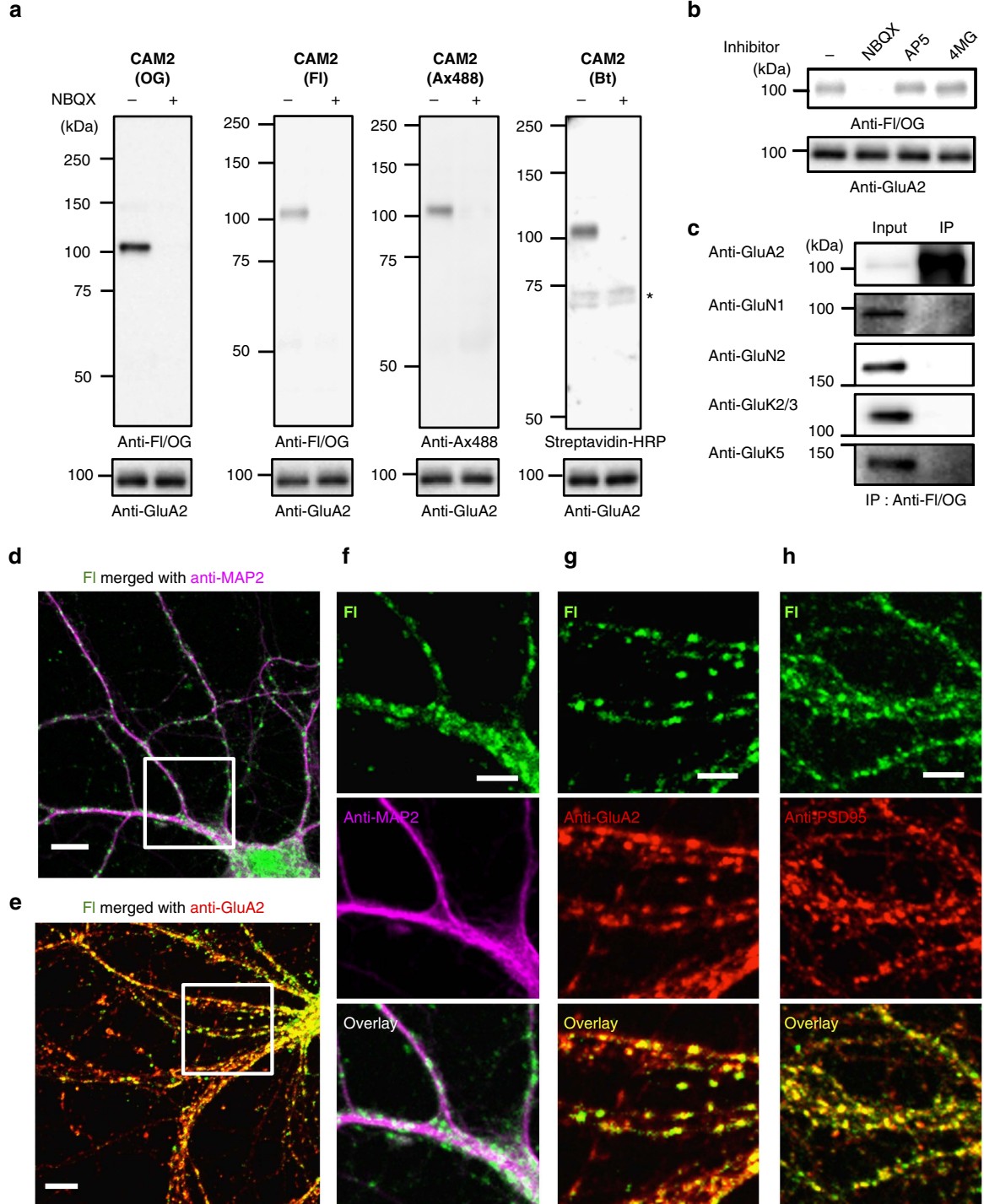

**Figure 3 | Chemical labelling of native AMPARs in cultured neurons.** (**a**) Western blot analyses of cultured neurons after labelling using CAM reagents. Cultured cortical neurons were treated with 1 μM of **CAM2(OG)**, **CAM2(Fl)**, **CAM2(Ax488)**, or **CAM2(Bt)** in the absence or presence of 10 μM NBQX in serum free Neurobasal medium. The cell lysates were analyzed by western blot using anti-Fl/OG, anti-Ax488, or anti-GluA2 antibody, or by biotin blotting using streptavidin-HRP. * indicates biotinylated proteins endogenously expressed in the neurons. (**b**) Effect of competitive antagonists for glutamate receptors on chemical labelling of native AMPARs in cultured neurons. Western blot analyses of cultured neurons after labelling using CAM reagents are shown. Cultured cortical neurons were treated with 1 μM of **CAM2(OG)** in the absence or presence of 10 μM NBQX, 10 μM AP5, or 10 μM (2S,4R)-4-methyl glutamate (4MG) to examine selective labelling of AMPARs among the ionotropic glutamate receptor family. (**c**) Analyses of labelled proteins in cultured neurons by immunoprecipitation using anti-Fl/OG antibody. Chemical labelling was conducted with the same procedure described in **a**. After lysis of labelled cultured neurons by **CAM2(Fl)**, the cell lysate was immunoprecipitated with anti-Fl/OG antibodies. The immunoprecipitates were analyzed by western blot using glutamate receptor-specific antibodies. (**d–h**) Confocal imaging of cultured neurons after labelling using CAM reagents. Cultured hippocampal neurons labelled with 1 μM **CAM2(Fl)** were fixed, permeabilized and immunostained using anti-MAP2 (in **d,f**), anti-GluA2 (in **e,g**) or anti-PSD95 antibody (in **h**). White square ROIs indicated in **d,e** are expanded in **f,g**, respectively. Scale bars, 10 μm (**d,e**) and 5 μm (**f–h**). Full blots for **b** and **c** are shown in Supplementary Fig. 22.

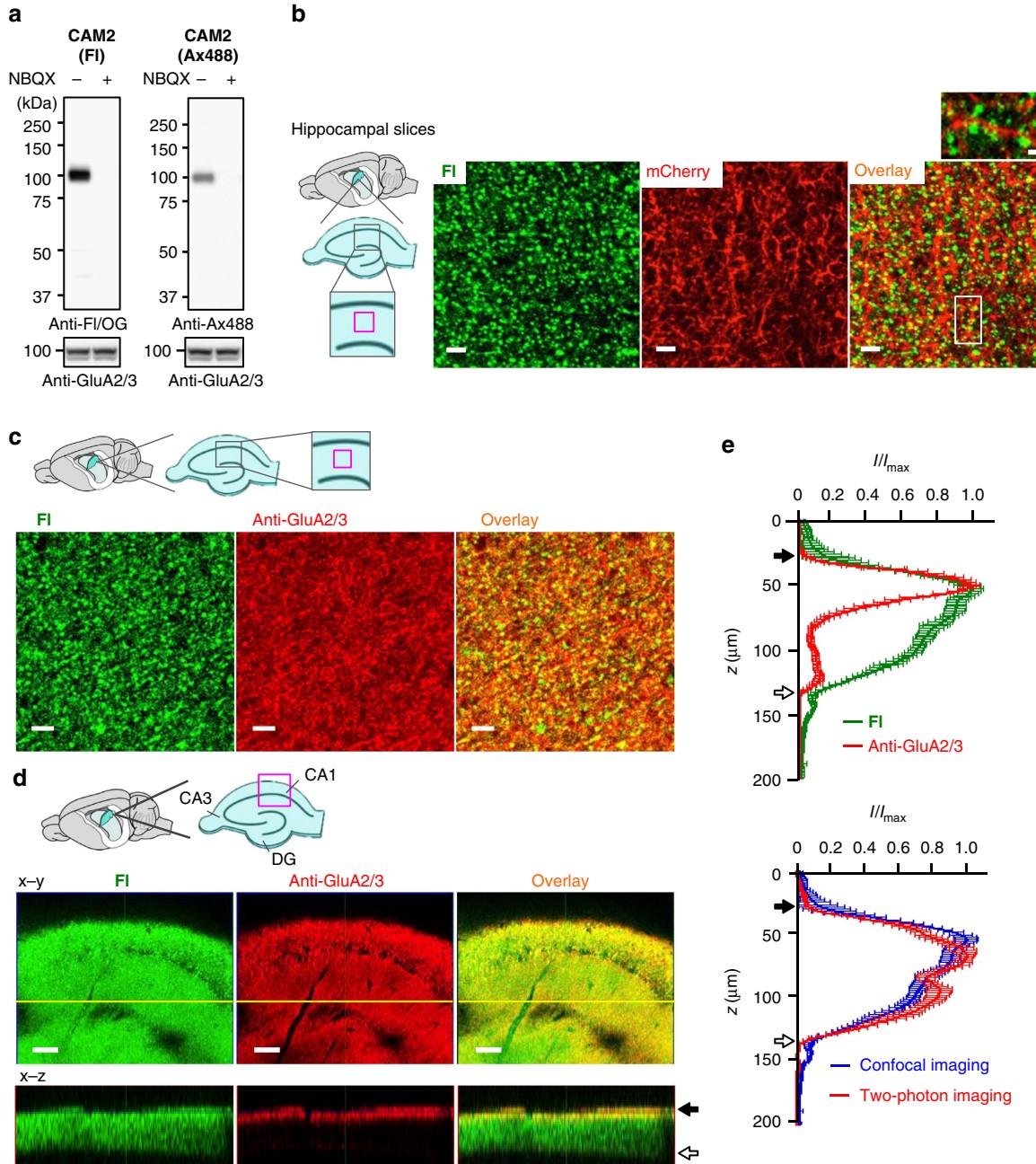

**Figure 4 | Chemical labelling of native AMPARs in brain slices.** (**a**) Western blot analyses of hippocampal slices after labelling using CAM reagents. Hippocampal slices were treated with 1 μM of **CAM2(FI)** or **CAM2(Ax488)** in the absence or presence of 10 μM NBQX in ACSF. The cell lysates were analysed by western blotting using anti-FI/OG, anti-Ax488, or anti-GluA2/3 antibody. (**b**) Confocal live imaging of labelled hippocampal slices with **CAM2(FI)**. To visualize neurons, hippocampal slices were acutely prepared from mice infected with lenti-virus encoding mCherry, and then treated with 1 μM of **CAM2(FI)** in ACSF for 1 h. The composite Z-stack images of the live hippocampal slices are shown. Scale bars, 5 μm. In left, imaged region is shown as a magenta square. In upper right, zoomed overlay image is shown. Scale bars, 2 μm. (**c**) Immunostaining of labelled hippocampal slices with **CAM2(FI)**. Hippocampal slices treated with 1 μM of **CAM2(FI)** in ACSF were fixed, permeabilized, and immunostained with anti-GluA2/3 antibody. Single plane confocal images of labelled slices immunostained with anti-GluA2/3 antibody are shown. Scale bars, 5 μm. (**d**) Single plane images (x–y) and ortho-images (x–z) of confocal Z-stacks of labelled hippocampal slices immunostained with anti-GluA2/3 antibody are shown. The yellow line indicates the region used for the x-z section. In the upper left, imaged region is shown as a magenta square. Closed or open arrow indicates top or bottom of the hippocampal slice, respectively. Scale bars, 100 μm. (**e**) Line profiling of Z-stack imaging shown in **d**. In top, comparison of line profiling of z-stack confocal imaging between labelled slice with **CAM2(FI)** (n = 12) and the slice immunostained with anti-GluA2/3 (n = 13) is shown. In bottom, line profiling of the labelled slices obtained by confocal (n = 12) or two-photon microscopy (n = 15) is shown. Gradual fluorescent decrease dependent on the depth of the slice was observed by confocal imaging. In contrast, the fluorescent change was not observed in two-photon imaging. These results suggest that the fluorescent change was caused not by decrease of labelling efficiency but by low transparency of the excitation light.

cerebellar slices. As shown in Fig. 5a,b, the amplitude and kinetics of excitatory postsynaptic currents (EPSCs) evoked by climbing fiber (CF) or parallel fiber (PF) stimuli were unaffected by chemical labelling with **CAM2(Fl)**. In addition, paired-pulse depression of CF-evoked EPSCs and paired-pulse facilitation of PF-evoked EPSCs were also unchanged, indicating that pre-synaptic glutamate release probability and lateral diffusion of postsynaptic AMPARs[47] are unaffected by this chemical labelling. Finally, the amplitude of miniature EPSCs, which represents the synaptic response to the release of glutamate from a single vesicle, was unaffected by **CAM2** labelling (Fig. 5c–e). Overall, these results indicate that **CAM2** reagents can be used to visualize native AMPARs under live conditions with no effect on synaptic function.

**Diffusion dynamics of endogenous AMPARs in live neurons.** The diffusion dynamics of cell-surface AMPARs have been studied by monitoring fluorescence recovery after photobleaching (FRAP) of SEP-AMPAR signals[48] and by single particle tracking of endogenous AMPARs labelled with fluorophore-tagged antibodies[49,50]. However, both methods have certain limitations, such as the large size of the antibodies, cross-linking of the receptors with antibodies[48], and pH sensitivity of the SEP[33]. To overcome these limitations, we examined diffusion dynamics of endogenous AMPARs labelled with **CAM2(Fl)** (**Fl**-AMPAR) using the FRAP method in cultured hippocampal neurons. After rapid photobleaching of a single puncta of the neurons, partial recovery of the fluorescence was observed (Fig. 6a,c). The recovery ratio (corresponding to the exchangeable component of **Fl**-AMPARs during this period) and diffusion coefficient were determined to be 16.2% and 0.090 $\mu m^2 s^{-1}$, respectively (Supplementary Table 1). These values are comparable to those reported previously by Chouqet's group[49] using single particle tracking methods under which the AMPAR mobility was unaffected by antibody cross-linking[50].

Next, we compared the diffusion dynamics of endo-genous **Fl**-AMPAR and exogenously expressed SEP-AMPAR.

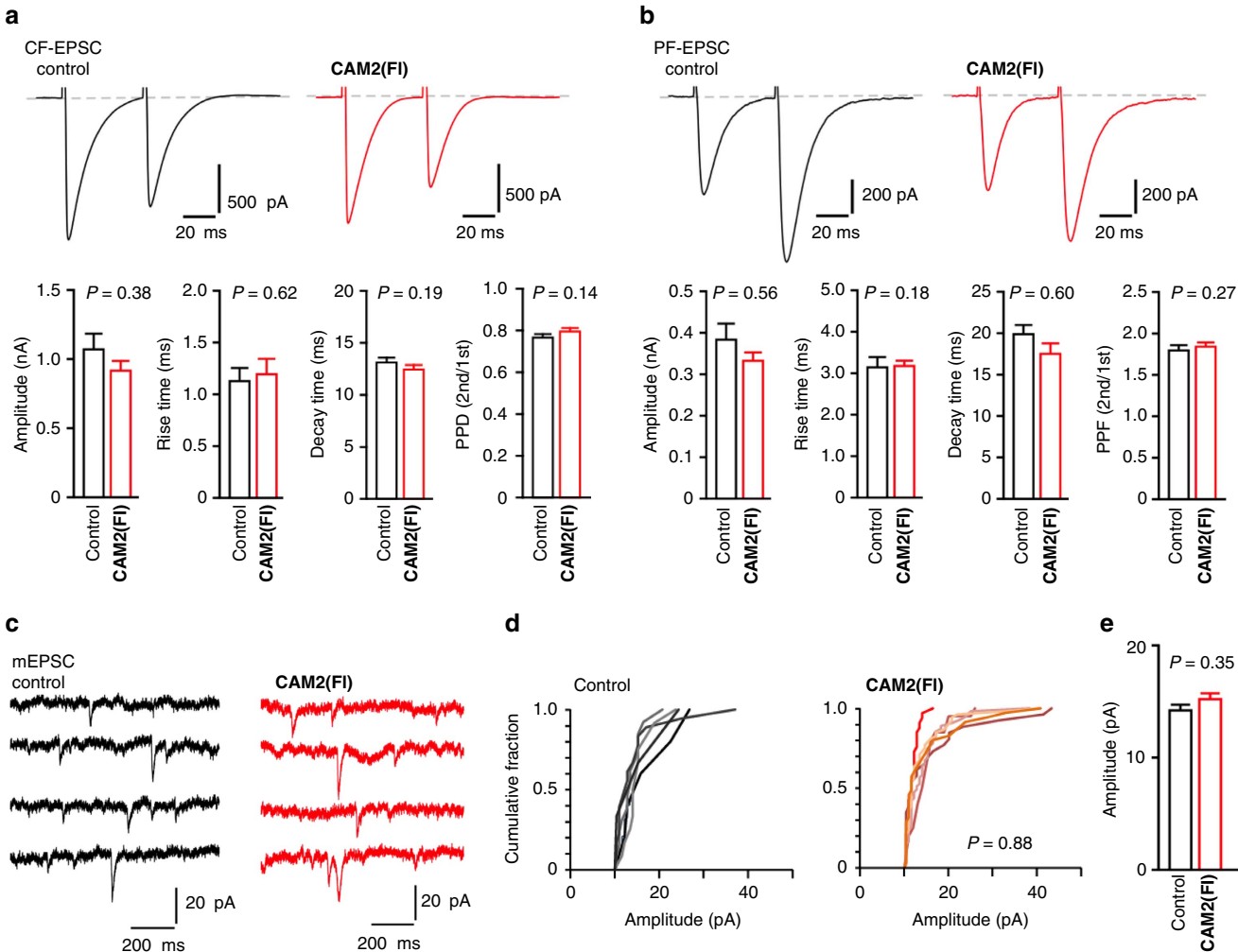

**Figure 5 | Chemical labelling does not affect AMPAR or neuronal functions in brain slices.** (**a,b**) Effects of chemical labelling using **CAM2** reagents on CF-EPSC (**a**) and PF-EPSC (**b**) in acutely prepared cerebellar slices. Cerebellar slices were treated with 1 μM of **CAM2(Fl)** and washed out three times with ACSF. Upper panels show representative EPSC traces recorded from Purkinje cells. Lower panels show peak amplitude, rise time, decay time, and paired-pulse ratio (PPD in **a** and PPF in **b**). The paired-pulse ratio of the EPSC amplitude was defined as the amplitude of the second EPSC divided by that of the first EPSC. n = 12 (control) or 15 (**CAM2(Fl)**) for CF-EPSC, n = 13 (control or **CAM2(Fl)**) for PF-EPSC. Data are represented as mean ± s.e.m. (**c–e**) Effects of chemical labelling using **CAM2** reagents on miniature EPSC (mEPSC) responses from Purkinje cells in acutely prepared cerebellar slices. (**c**) Representative mEPSC traces recorded from Purkinje cells. (**d**) Cumulative probability plot showing the distribution of mEPSC amplitude. (**e**) Averaged amplitude of mEPSC. n = 12 (control) or 15 (**CAM2(Fl)**). Data points are mean ± s.e.m. Kolmogorov–Smirnov test in **d** and Mann–Whitney U-test in **e** indicate that significant differences were not observed with or without the chemical labelling.

Interestingly, the exchangeable component of SEP-AMPAR (54.1%) was substantially higher than the value for **Fl**-AMPAR (16.2%) (Fig. 6b,c and Supplementary Table 1). In addition, the

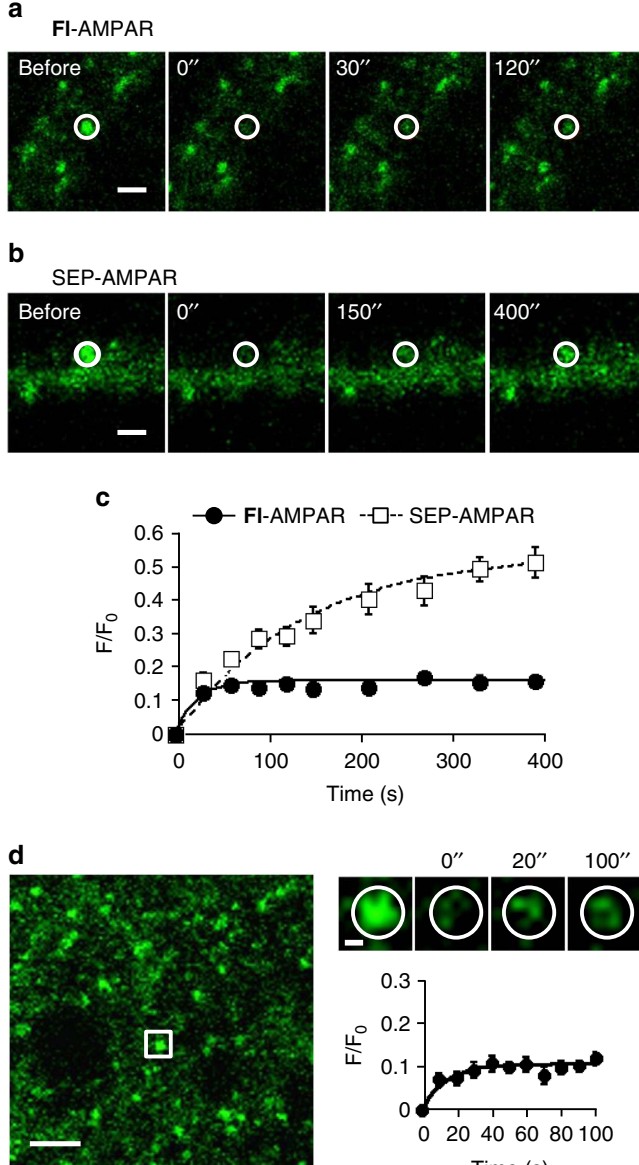

**a** FI-AMPAR

Before | 0″ | 30″ | 120″

**b** SEP-AMPAR

Before | 0″ | 150″ | 400″

**c**

● **FI**-AMPAR  □ **SEP-AMPAR**

$F/F_0$ vs Time (s)

**d**

0″ | 20″ | 100″

$F/F_0$ vs Time (s)

**Figure 6 | FRAP analyses for diffusion dynamics of AMPARs in cultured neurons or brain slices.** (**a–c**) FRAP analyses of **Fl**-AMPARs or SEP-AMPARs in cultured hippocampal neurons. In **a**, representative FRAP images of endogenous AMPARs labelled with **CAM2(Fl)** (**Fl**-AMPAR) are shown. Hippocampal neurons were treated with 1 μM of **CAM2(Fl)** for 1 h at 17 °C in ACSF. Labelling yield of **CAM2** to surface-exposed GluA2 in this condition was determined as 9.6 ± 0.9% (Supplementary Fig. 20). Bleached regions are indicated by white circle. Scale bars, 2 μm. In **b**, representative FRAP images of exogenously expressed SEP-AMPARs (SEP-GluA2) are shown. In **c**, averaged FRAP curves for **Fl**-AMPARs ($n = 16$) and SEP-AMPARs ($n = 8$) are shown. (**d**) FRAP analyses of **Fl**-AMPARs in hippocampal slices. Hippocampal slices were treated with 1 μM of **CAM2(Fl)** in ACSF. In left, a confocal image of labelled slice is shown. Scale bar, 2 μm. In upper right, representative FRAP images of a puncta are shown. The white square ROI in the left image is zoomed, and the bleached region is indicated by white circle. Scale bar, 500 nm. In lower right, averaged FRAP curve is shown ($n = 16$). Data are represented as mean ± s.e.m.

recovery of SEP-AMPAR fluorescence ($t_{1/2} = 95.1$ s) was slower relative to that for endogenous **Fl**-AMPAR ($t_{1/2} = 15.6$ s) (Supplementary Table 1). To test the possibility that such differences were caused by overexpression of AMPARs in the analysis of SEP-AMPARs, we next labelled hippocampal neurons expressing SEP-AMPAR with **CAM2(Ax647)**. The dual colour FRAP analysis revealed that the diffusion dynamics of **Ax647**-AMPARs in neurons expressing SEP-AMPARs were comparable to that of **Fl**-AMPARs obtained in non-transfected neurons (Supplementary Fig. 21). These results indicate that high recovery ratio and slow kinetics of the SEP-AMPAR could not be ascribed to overexpression of AMPARs. A plausible explanation is the involvement of other trafficking processes, such as those from intracellular acidic organelles to the surface owing to the pH-sensitivity of SEP (for details, see Discussion).

Because antibodies are less able to permeate into tissues, single particle tracking techniques are only applicable to cultured neurons. Taking advantage of the high penetration and labelling capability of our **CAM2** reagents, we next analyzed the diffusion dynamics of endogenous AMPARs in brain slices. The recovery ratio and diffusion coefficient of endogenous **Fl**-AMPARs in freshly prepared hippocampal slices was determined to be 10.5% and 0.095 $\mu m^2 s^{-1}$, respectively (Fig. 6d and Supplementary Table 1). Although marginally lower, this recovery ratio was comparable to that obtained using endogenous **Fl**-AMPARs in cultured neurons (Fig. 6c). To the best of our knowledge, this is the first report showing constitutive diffusion dynamics of endogenous AMPARs in three-dimensional brain slices.

## Discussion

In this work, we described the development of useful chemical tools for the selective labelling and imaging of endogenous AMPARs in live cultured neurons and brain tissues. Using **CAM2** reagents, this method enables covalent modification of AMPARs in live neurons with small chemical probes such as Oregon green, fluorescein, Alexa488, Alexa647 and biotin in close proximity to the AMPAR ligand-binding domain with no need for any genetic manipulation. We anticipate that biotin labelling using **CAM2(Bt)** followed by incubation with gold nanoparticle-streptavidin conjugate will be applicable to electron microscopy analyses. Similarly, **CAM2(Ax647)** could be exploited for chemical labelling of AMPARs in super-resolution imaging of native AMPARs using the STORM or PALM techniques. With respect to the study of receptor dynamics, **CAM2(CypHer)** can be used for endocytosis or recycling assays of native AMPARs. For pulse-chase analyses, chemical labelling using two **CAM2** reagents with different emission wavelengths (e.g., any combination of **CAM2(Ax488)**, **CAM2(Cy3.5)** and **CAM2(Cy5.5)**) should prove beneficial. Moreover, highly photo-stable **CAM2** reagents could be exploited for single particle tracking studies. These putative applications highlight the exciting potential of our method.

It is generally known that glial cells including astrocytes tightly wrap synapses from the post-synaptic side in brain tissues[51]. In the hippocampus of mature animals, approximately 57% of synapses are sheathed. Notably, in the cerebellum, almost all mature cerebellar excitatory synapses on Purkinje cells are entirely covered by Bergmann glia. Thus, large molecules including antibodies often cannot reach synaptic receptors in live brain tissues owing to this low accessibility. Here we successfully labelled and visualized AMPARs in intact hippocampal and cerebellar slices using small molecules. Notably, cerebellar molecular layers, which are enriched in PF and CF synapses on the dendritic tree of Purkinje cells, were clearly visualized (Supplementary Fig. 19). This demonstrates the high penetrability of **CAM2** reagents into the narrow synaptic

clefts (15–20 nm) in brain tissues. Thus, our method appears suitable for probing AMPARs located in deep brain areas that cannot be accessed using conventional methods.

Small chemical probes for visualizing cell types or subcellular components under live conditions have served as powerful tools for characterizing neuronal function and dysfunction in brain tissues. For instance, sulforhodamine 101 has been utilized as an astroglial marker in live tissues[52]. Recently, a new fluorescent chemical probe, NeuO has been developed to selectively visualize neurons[53]. Membrane-staining dyes such as DiI and DiO have been widely employed as anterograde and retrograde neuronal tracers[54], and stryryl dyes such as FM1-43 are useful for pre-synaptic vesicle recycling studies[55]. In addition, fluorescent false neurotransmitters have been applied to the study of neurotransmitter release at individual presynaptic terminals[56]. However, with respect to visualization of postsynaptic events, no small chemical probes have yet been developed that can achieve molecular level resolution. We here demonstrated that AMPARs labelled with **CAM2** reagents merged well with excitatory postsynaptic markers. Thus, **CAM2** reagents represent a new paradigm for visualizing AMPAR-expressing excitatory postsynapses in cultured neurons and neuronal tissues.

We found that the recovery ratio was higher with SEP-AMPARs than with **Fl**-AMPARs labelled with **CAM2** reagents. SEP labelling is a conventional method in membrane trafficking (endocytosis and/or exocytosis) studies, where it exploits the high pH-sensitivity of SEP to produce high fluorescence at the cellular surface while minimizing signals from the acidic intracellular organelles[2–4]. Such pH responsiveness results in endosomes containing SEP-AMPARs that are less susceptible to photobleaching owing to weakened fluorescence under acidic conditions. Thus, a larger recovery fraction would be anticipated, as it would represent not only lateral diffusion but also exocytosis of unbleached intracellular SEP-AMPARs. The time scale of exocytosis determined by direct observation using SEP-AMPARs[57] is consistently in the same range as the slow recovery rate obtained here. In addition, reversible photoswitching of SEP[58,59] might have occurred during the fluorescent recovery of the SEP-AMPARs, which would cause additional complicating factors. Moreover, exogenously expressed AMPARs may behave differently than endogenous AMPARs due to the formation of nonphysiological tetramers. Thus, careful consideration would be needed to assess the diffusion dynamics of SEP-AMPARs.

Trafficking of AMPARs is dynamically regulated during synaptic plasticity, and the number of postsynaptic AMPARs is tightly regulated by the balance between insertion and internalization of receptors. To visualize these events in live neurons, SEP-tagged AMPARs have been widely utilized[3,4]. However, it is recently pointed out that pH changes in the intracellular pools could affect SEP-AMPARs fluorescent signals[33]. Since **CAM2** predominantly labelled surface-exposed AMPARs, it is expected to serve as a simple tool to monitor the trafficking of cell surface AMPARs in live neurons. Moreover, abnormalities of AMPAR trafficking are strongly implicated in many neurodevelopmental and neuropsychiatric disorders, such as epilepsy, autism, depression and schizophrenia[60]. Therefore, this **CAM2**-based method is a powerful and versatile tool for characterizing the physiological and pathophysiological status of AMPAR trafficking in live neurons.

## Methods

**Synthesis.** All synthetic procedures and compound characterizations are described in Supplementary Methods.

**General methods for biochemical and biological experiments.** SDS–PAGE and western blotting were carried out using a Bio-Rad Mini-Protean III electrophoresis apparatus. Chemiluminescent signals generated with Chemi-Lumi One (nacalai

tesque) or ECL Prime (GE Healthcare) were detected with an LAS4000 imaging system (Fuji Film). All experiment procedures were performed in accordance with the National Institute of Health Guide for the Care and Use of Laboratory Animals and approved by the Institutional Animal Use Committees of Kyoto University or Keio University.

**Construction of expression plasmids.** Utilizing a PCR method, cDNA encoding an HA tag was added to the 5′ end (immediately following the signal sequence) of mouse GluA2$^{flop}$(R), GluA2$^{flip}$(Q), or GluA2$^{flop}$(Q), GluA1$^{flip}$(Q), GluA3$^{flip}$(Q) or GluA4$^{flip}$(Q) cDNA. All PCR-amplified DNAs were confirmed by DNA sequence analyses. These cDNAs were subcloned into the expression vector, pCAGGS (kindly provided by Dr J. Miyazaki, Osaka University, Osaka, Japan). To obtain SNAP-AMPAR (SNAP-GluA2$^{flop}$(R)), cDNA encoding SNAP-tag was amplified from pSNAP$_f$ vector (NEB). After DNA sequence analyses, the SNAP-tag cDNA was inserted between HA-tag and GluA2$^{flop}$(R) in the pCAGGS expression vector.

**Chemical labelling of AMPARs in HEK293T cells.** HEK293T cells (ATCC) were cultured in Dulbecco's modified Eagle's medium (DMEM)-Glutamax (Invitrogen) supplemented with 10% fetal bovine serum (Invitrogen), penicillin (100 units ml$^{-1}$), streptomycin (100 µg ml$^{-1}$), and amphotericin B (250 ng ml$^{-1}$), and incubated in a 5% $CO_2$ humidified chamber at 37 °C. Cells were transfected with a plasmid encoding flop form of RNA-edited GluA2 (GluA2$^{flop}$(R)) or the control vector using the lipofectamine 2000 (Invitrogen) and subjected to labelling experiments after 36 h of the transfection. For chemical labelling, the cells expressing GluA2 were washed with serum-free DMEM-Glutamax (25 mM HEPES), and treated with 2 µM labelling reagents (**CAM1(OG)**, **CAM2(OG)**, or **CAM3(OG)**) in the absence or presence of 50 µM NBQX in the serum free medium at 17 °C for 4 h to suppress internalization of AMPARs[37].

For live imaging experiments, HEK293T cells were co-transfected with GluA2 and mCherry-F as a transfection marker. After chemical labelling as described above, the cells were washed 3 times with ice-cold HBS (20 mM HEPES, 107 mM NaCl, 6 mM KCl, 2 mM $CaCl_2$, and 1.2 mM $MgSO_4$ at pH 7.4). Cell imaging was performed in a confocal microscopy (FV1000, IX81, Olympus) equipped with a 60 ×, numerical aperture (NA) = 1.35 oil objective. Fluorescence images were acquired using a 488 nm line of an argon laser for excitation of **OG** and a HeNe Green laser for excitation of mCherry-F.

For western blot analysis, after chemical labelling, cells were washed 3 times with ice-cold HBS, lysed with radio immunoprecipitation assay (RIPA) buffer containing 1% protease inhibitor cocktail set III (Calbiochem), and mixed with a quarter volume of 5 × SDS–PAGE loading buffer containing 250 mM DTT. The samples were applied to SDS-PAGE and electrotransferred onto immune-blot polyvinylidene fluoride (PVDF) membranes (Bio-Rad), followed by blocking with 5% nonfat dry milk in Tris-buffered saline (TBS) containing 0.05% Tween (Sigma-Aldrich). The **OG**-labelled GluA2 was detected by chemiluminescence analysis using rabbit anti-fluorescein antibody (Abcam, ab19491, × 1,000) and anti-rabbit IgG-HRP conjugate (Santa Cruz, sc-2004, × 3,000). The immunodetection of GluA2 was performed with a rabbit anti-GluA2/3 antibody (Millipore, 07-598, × 3,000) and anti-rabbit IgG-HRP conjugate (Santa Cruz, sc-2004 × 3,000). The signals were developed with Chemi-Lumi One (Nacalai tesque) or ECL Prime Western Blotting Detection Reagent (GE Healthcare) and detected with Imagequant LAS4000 (GE Healthcare).

**Chemical labelling of ligand binding domain of GluA2 (S1S2J).** Production, refolding and purification of a ligand binding domain of GluA2 (S1S2J) were carried out as previously described[61] (The expression plasmid was kindly gifted from Professor Gouaux). Three µM S1S2J was incubated with 6 µM **CAM2(OG)** with or without 100 µM NBQX inhibitor in the 20 mM HEPES buffer (100 mM NaCl, pH 7.2) at 17 °C. At indicated time points, each sample was mixed with an equal volume of 2 × SDS − PAGE loading buffer (125 mM Tris-HCl, 100 mM DTT, 4% SDS, 20% glycerol, and 0.01% bromophenol blue (BPB), pH 6.8). The samples were subjected to SDS-PAGE, and **OG**-labelled S1S2J was detected and analyzed by an in-gel fluorescence imaging system (LAS4000). After fluorescence imaging, the gel was stained by Imperial Protein Stain (Themo Scientific).

**Identification of labelling site.** Three µM S1S2J was incubated with 20 µM **CAM2(OG)** in the 20 mM HEPES buffer (100 mM NaCl, pH7.2) at 17 °C for 62 h. After adding 1 mM glutamate to dissociate excess **CAM2(OG)** and the resultant ligand moiety from S1S2J, the **OG**-labelled S1S2J was purified by size-exclusion chromatography using a TOYOPEARL HW-40F column (Tosoh Corporation). Two M Urea and trypsin (trypsin/substrate ratio = 1/30 (w/w)) were added to the resulting solution. After incubation at 37 °C for 14 h, the digested samples were applied to RP-HPLC. The trypsin-digested labelled fragments were analyzed by MALDI-TOF MS and MALDI-TOF MS/MS (Autoflex II, Bruker Daltonics).

**Reciprocal immunoblot analyses of labelled GluA2.** HEK293T cells expressing GluA2 were washed with serum-free DMEM-Glutamax (25 mM HEPES) and treated with 2 µM **CAM2(OG)** in serum free DMEM-Glutamax (25 mM HEPES) at 17 °C for 4 h. The following lysis, electrophoresis and blotting processes were

similarly performed as described above. The **OG**-labelled GluA2 was detected by chemiluminescence analysis using rabbit anti-Fl/OG antibody (Abcam, $\times$ 1,000) and anti-rabbit IgG-HRP conjugate (Santa Cruz, $\times$ 3,000). After membrane was stripped by stripping solution (25 mM glycine, 1% SDS, pH 2.0), the immunodetection of GluA2 was performed with a mouse anti-GluA2 antibody (Millipore, MAB397, $\times$ 3,000) and anti-mouse IgG-HRP conjugate (Santa Cruz, sc-2005, $\times$ 3,000).

**$Ca^{2+}$ response analysis of labelled AMPARs in HEK293T cells.** HEK293T cells transfected with a flip form of $Ca^{2+}$-permeable GluA2 (GluA2$^{flip}$(Q)) and DsRed (transfection marker) were plated on glass coverslips. 34 h after transfection, the cells were treated with 1 $\mu$M Fura2-AM ($Ca^{2+}$ indicator, Dojindo) in culture medium at 37 °C for 20 min in a 5% $CO_2$ humidified chamber. The cells were washed with serum free DMEM-Glutamax (25 mM HEPES) and treated with 3 $\mu$M **CAM2(OG)** in serum free DMEM-Glutamax (25 mM HEPES) at 17 °C for 4 h. The coverslips were placed on the stage of a fluorescent microscopy (IX71, Olympus) equipped with a 20 $\times$, numerical aperture (NA) = 0.75 objective and continually perfused with HBS (20 mM HEPES, 107 mM NaCl, 6 mM KCl, 2 mM $CaCl_2$, 1.2 mM $MgSO_4$ and 11.5 mM glucose at pH 7.4). Cells were imaged using an AQUACOSMOS system (Hamamatsu photonics). The Cells expressing DsRed were marked and excited at 340 and 380 nm with emissions collected at 520 nm at 5 s intervals. One-hundred $\mu$M Cyclothiazide (CTZ) and 100 $\mu$M glutamate were applied during periods indicated by shaded bars, and $[Ca^{2+}]_i$ changes (340/380 nm excitation fluorescence ratio; ratio(ex340/ex380)) evoked by 100 $\mu$M glutamate were measured.

**Outside-out recording in HEK293 cells.** HEK293 cells (ATCC) transfected with GluA2$^{flop}$(Q) as previously[62] were labelled with 10 $\mu$M **CAM2(Ax488)** for 4 h at 17 °C. Transfected cells were identified by the fluorescence of EGFP. Outside-out patch-clamp recordings were made at room temperature with an Axopatch 200B (Molecular Device). Thin-wall borosilicate glass pipettes (World Precision Instruments) had resistances of 4–8 M$\Omega$ when filled with an intracellular solution composed of (in mM): 150 CsCl, 4 $MgCl_2$, 10 HEPES, 0.4 EGTA, 4 $Na_2$ATP, 1 $Na_2$GTP and 5 QX-314 (pH 7.3). The HEK293 cells were superfused with an extracellular solution composed of (in mM): 150 NaCl, 5 KCl, 2 $CaCl_2$, 20 HEPES and 20 mM D-glucose (pH 7.3). Outside-out membrane patches from transfected HEK293 cells were voltage-clamped at – 60 mV. Responses to agonists were low-pass filtered at 2 kHz with an 8-pole Bessel filter and digitized at 50 kHz. Solutions were gravity fed into each lumen of theta glass tubing (tip diameter, *ca.* 300 $\mu$m; Harvard Apparatus). The patch was positioned near the interface formed between continuously flowing control and glutamate-containing solutions. Solution exchange was made by rapidly moving the theta glass with a Piezo translator (LSS-3100, Burleigh Instruments). This system typically permitted solution exchange in < 200 $\mu$s (20–80% rise time) as determined by measurements of open-tip junction currents after disruption of the patch at the end of every experiment.

**Preparation of primary cortical neuronal culture.** Cerebral cortices from 16-day-old Sprague Dawley rat embryos were aseptically dissociated and digested with 0.25 w/v% trypsin (Nacalai tesque) for 20 min. After centrifugation, the cells were re-suspended in Neurobasal medium supplemented with 2% NS21 supplement[63], 0.5 mM glutamine (Invitrogen), penicillin (100 units per ml) and streptomycin (100 $\mu$g ml$^{-1}$), and were plated on 24-well plate (Falcon) coated with poly-D-lysine (Sigma-Aldrich), and maintained at 37 °C in a humidified atmosphere of 95% air and 5% $CO_2$. The cultured medium was exchanges every 3 days, and the neurons were used at 11–14 days *in vitro* (DIV). **CAM2** reagents tend to be incorporated into dead cells. Thus, we utilized NS21, a serum-free supplement for our neuronal culture.

**Chemical labelling of endogenous AMPARs in cortical neurons.** The cultured cortical neurons were washed with serum free Neurobasal medium (10 mM HEPES), treated with 1 $\mu$M labelling reagents (**CAM2(OG)**, **CAM2(Fl)**, **CAM2(Ax488)** or **CAM2(Bt)**) in serum free serum free Neurobasal medium (10 mM HEPES) and incubated at 17 °C for 4 h to suppress internalization of AMPARs[37]. As a control experiment, the labelling was conducted in the presence of 10 $\mu$M NBQX, 10 $\mu$M AP5, 10 $\mu$M (2 S,4 R)-4-methyl glutamate. Then, the dishes were washed with ice-cold HBS three times and lysed with RIPA buffer containing 1% protease inhibitor cocktail set III. After mixing with a quarter volume of 5 $\times$ SDS–PAGE loading buffer containing 250 mM DTT, the following electrophoresis processes were similarly performed as described above. The labelled GluA2 was analyzed using an anti-fluorescein antibody for **OG**-AMPAR and **Fl**-AMPAR (abcam, ab19491, $\times$ 1,000), anti-Alexa488 antibody for **Ax488**-AMPAR (Invitrogen, A11094, $\times$ 1,000) or avidin-HRP conjugate for **Bt**-AMPAR (Invitrogen, S911, $\times$ 3,000). For western blot analysis of glutamate receptors, a rabbit anti-GluA2 antibody (abcam, ab20673, $\times$ 1,000), a mouse anti-GluN1 antibody (BD Pharmingen, 556308, $\times$ 1,000), a rabbit anti-GluN2 antibody (Cell Signaling, D15B3, $\times$ 1,000), a rabbit anti-GluK2/3 (Millipore, 04-921, $\times$ 1,000), and a rabbit anti-GluK5 (Millipore, 06-315, $\times$ 1,000) were utilized. For immunoprecipitation, an anti-fluorescein antibody (Abcam, ab19491) was utilized.

The signals were developed with Chemi-Lumi One or ECL-Prime and detected with LAS4000.

**Preparation of primary hippocampal neuronal culture.** Hippocampal neuronal culture was performed as previously reported with minor modifications[64]. Hippocampi from 18-day-old Sprague Dawley rat embryos were aseptically dissociated and digested with 0.25 w/v% trypsin (Nacalai tesque) for 20 min. After centrifugation, the cells were re-suspended in Neurobasal medium supplemented with 2% NS21 supplement, 0.5 mM glutamine (Invitrogen), penicillin (100 units per ml) and streptomycin (100 $\mu$g ml$^{-1}$), and were plated on glass coverslip (Matsunami) or glass-bottom dish (Matsunami) coated with poly-D-lysine (Sigma-Aldrich) and laminine (Sigma-Aldrich), and maintained at 37 °C in a humidified atmosphere of 95% air and 5% $CO_2$. The cultured medium was exchanges every 7 days, and the neurons were used at 16–22 DIV. For transfection, after 6–22 DIV, the neurons were transiently transfected with plasmids using Lipofectamine 2,000 according to the manufacturer's instruction. **CAM2** reagents tend to be incorporated into dead cells. Thus, we utilized NS21, a serum-free supplement for our neuronal culture.

**Immunostaining of cultured neurons after labelling.** Hippocampal cells cultured on glass coverslips were washed with serum free Neurobasal medium (10 mM HEPES), and treated with 1 $\mu$M **CAM2(Ax488)** in serum-free Neurobasal medium (10 mM HEPES) at 17 °C for 4 h. Then, the cells after labelling were washed with HBS buffer and fixed with 4% paraformaldehyde at room temperature (r.t.) for 30 min and washed with PBS buffer. This was followed by permeabilization with PBS containing 0.2% triton X-100 at r.t. for 10 min and blocking with PBS containing 10% normal goat serum for 1 h. After blocking, primary antibody in PBS buffer containing 5% normal goat serum was added and incubated at 4 °C for 12 h. Secondary antibody in PBS buffer containing 5% normal goat serum and was added and incubated at r.t. for 1 h. Used primary antibodies were as follows: mouse anti-GluA2 (Millipore, MAB397, $\times$ 300), mouse anti-PSD95 (abcam, ab2727, $\times$ 300) and rabbit anti-MAP2 (Millipore, AB5622, $\times$ 300). Secondary antibodies were conjugated to Alexa546 or Alexa633 fluorophores (Invitrogen, A11071, A11018, A21050, or A21070, $\times$ 1,000). Cell imaging was performed with a confocal microscopy (LSM710, Axio Observer.Z1, ZEISS) equipped with a 63 $\times$, numerical aperture (NA) = 1.40 oil objective. Fluorescence images were acquired using a 488 nm line of an argon laser for excitation of **Ax488**, DPPS laser for excitation of Alexa546 and HeNe laser for excitation of Alexa633.

**Surface immunostaining of GluA2 in hippocampal neuron.** Hippocampal cells cultured on glass coverslips were washed with serum free Neurobasal medium (10 mM HEPES), and treated with 1 $\mu$M **CAM2(Fl)** in serum free Neurobasal medium (10 mM HEPES) at 17 °C for 4 h. Then, the cells after labelling were washed with HBS buffer and fixed with 4% paraformaldehyde at r.t. for 30 min and washed with PBS buffer. This was followed by blocking with PBS containing 10% normal goat serum for 1 h. After blocking, mouse anti-GluA2 antibody (Millipore, MAB397, x300) and rabbit anti-MAP2 (Millipore, AB5622, $\times$ 300) in PBS buffer containing 5% normal goat serum was added and incubated at 4 °C for 12 h. Secondary antibody conjugated to Alexa546 or Alexa633 (Invitrogen, A11071 or A21070) in PBS buffer containing 5% normal goat serum and was added and incubated at r.t. for 1 h. Cell imaging was performed as described above.

**Trypan blue treatment to labelled neurons.** HEK293T cells expressing GluA2$^{flip}$(Q) on glass coverslip were washed with serum free DMEM-Glutamax (25 mM HEPES) and treated with 2 $\mu$M of **CAM2(Ax488)** at 17 °C for 4 h. For promoting labelled GluA2 internalization, the cells were washed 3 times with serum-free DMEM and incubated at 37 °C for 2 h in DMEM (10% fetal bovine serum). After the labelling procedure, the glass coverslip was placed on the stage of a fluorescent microscopy (IX83-ZDC2, Olypus) equipped with a 60 $\times$, numerial aperture (NA) = 1.3 objective and continually perfused with HBS. Cells were imaged using a HCImage system (Hamamatsu photonics). 0.4% trypan blue (TB) solution in PBS were applied for 5 min. Fluorescence images before and after TB treatment were aquired with 488 nm excitation. For the neuron experiment, cultured hippocampal neurons were washed with serum free Neurobasal medium (10 mM HEPES) and treated with 1 $\mu$M of **CAM2(Ax488)** at 17 °C for 4 h. The following imaging processes were similarly performed as described above.

**Live imaging of labelled AMPARs in cultured neurons.** Cultured hippocampal neurons were washed with ACSF solution (120 mM NaCl, 5 mM KCl, 2 mM $CaCl_2$, 1.2 mM $MgCl_2$, 30 mM D-glucose, 25 mM HEPES and 1 $\mu$M tetrodotoxin, pH 7.4) and treated with 1 $\mu$M **CAM2(Fl)** in ACSF solution at 17 °C for 1 h. After labelling procedure, the cells were placed on the stage of confocal microscopy (LSM710, Axio Observer.Z1, ZEISS) equipped with Definite Focus (ZEISS) (for compensating focus drifts) and continually perfused with ACSF solution. Continuous perfusion is required for removal of excess **CAM2** reagents and the resultant ligand moiety. Fluorescence was excited using 63 $\times$ oil objective (NA = 1.40) by a 488 nm line of an argon laser. To induce chem-LTD, NMDA solution (50 $\mu$M in ACSF) was perfused for 10 min.

**Chemical labelling and western blot analysis of brain slices.** Hippocampal or cerebellar slices (200 or 250 μm thickness) were prepared from P14–21 ICR mice. The hippocampal slices were treated with 1 μM **CAM2(Fl)** or **CAM2(Ax488)** in ACSF solution (125 mM NaCl, 2.5 mM KCl, 2 mM CaCl$_2$, 1 mM MgCl$_2$, 1.25 mM NaH$_2$PO$_4$, 26 mM NaHCO$_3$, 10 mM D-glucose and 100 μM Picrotoxin) at r.t. for 1 h under 95% O$_2$/5% CO$_2$. As a control experiment, the labelling was conducted in the presence of 10 μM NBQX. Then, the slices after labelling were washed with ACSF solution three times and lysed with RIPA buffer containing 1% protease inhibitor cocktail set III. After mixing with the a quarter volume of 5 × SDS–PAGE loading buffer containing 250 mM DTT, the following electrophoresis and western blot analyses were similarly performed as described in 'Chemical labelling of endogenous AMPARs in cortical neurons'.

**Immunostaining of labelled hippocampal or cerebellar slices.** After the labelling procedure, the hippocampal or cerebellar slices were fixed with 4% paraformaldehyde at r.t. for 3 h. This was followed by permeabilization and blocking with PBS containing 2% bovine serum albumin, 2% normal goat serum 0.2% triton for 30 min. If these steps are insufficient, fluorescence may be remained in cell bodies. Then, primary antibody reaction was conducted with a rabbit anti-GluA2/3 (Millipore, AB1506, × 500) or a mouse anti-MAP2 (Millipore, MAB378, × 1,000) in PBS at r.t. for 12 h. Secondary antibody reaction was conducted with an Alexa546 anti-rabbit (Invitrogen, A11071, × 1,000) or Dylite650 anti-mouse (abcam, ab96784, × 1,000) in PBS at r.t. for 1 h. The z-stack images of the stained slices were taken using a confocal microscopy (LSM710, Axio Observer.Z1, ZEISS) or two-photon microscopy (LSM780 NLO, Axio Observer.Z1, ZEISS) with 10x objective (NA = 0.45) or 63 × oil objective (NA = 1.40). Fluorescence images were acquired using a 488 nm line of an argon laser for excitation of Fluorescein, DPPS laser for excitation of Alexa546 and HeNe laser for excitation of Dylite650. **CAM2** reagents tend to be incorporated into dead cells. Thus, acutely prepared brain slices should be maintained in a healthy state for visualizing surface-exposed AMPARs.

**Recombinant lentivirus and in vivo injection.** Production of recombinant lentivirus and in vivo injection were performed as previously reported[65]. In brief, to produce the lentivirus vectors, the plasmids for the VSV-G (G glycoprotein of vesicular stomatitis virus)-psuedotyped vectors were used. pCL36 carrying a mCherry gene was transfected to HEK293T cells together with helper plasmids by the calcium phosphate method. Eighteen hours after transfection, cells were washed with fresh culture medium (DMEM with 10% FBS) and allowed to produce virus particles for 24 h. The culture supernatant was gently applied over the 20% sucrose solution and centrifuged at 6,000g for 16 h at 4 °C for concentrating the lentivirus vector at the level of 10$^9$ titer unit (TU).

For in vivo virus infection, under deep anaesthesia with an intraperitoneal injection of ketamine/ xylazine (80/20 mg kg$^{-1}$, Sigma), the solution containing the mCherry coding lentivirus (2.5 μl; titer, 1.0 × 10$^9$ TU ml$^{-1}$) was injected into the CA1 region of the dorsal hippocampus of ICR mice aged postnatal day 14–22 stereotaxically (2.0–2.3 mm posterior to the Bregma, 1.5–2.0 mm lateral to the midline and 1.5–2.0 mm ventral from the pial surface). Twenty-four hours after injection, the infected mice were subjected to transverse hippocampal slices.

**Whole-cell recording from Purkinje cells in brain slices.** Parasagittal cerebellar slices (200-μm thick) were prepared from ICR mice (postnatal day 14–21) as described previously[65,66], and then the slices were treated with 1 μM **CAM2(Fl)** in ACSF for 1 h at room temperature under 95% O$_2$/5% CO$_2$. Whole-cell patch-clamp recordings were made from visually identified Purkinje cells using a 60 × water-immersion objective attached to an upright microscope (BX51WI, Olympus) at r.t. The solution used for recording consisted of the following (in mM): 125 NaCl, 2.5 KCl, 2 CaCl$_2$, 1 MgCl$_2$, 1.25 NaH$_2$PO$_4$, 26 NaHCO$_3$ and 10 D-glucose, bubbled continuously with a mixture of 95% O$_2$ and 5% CO$_2$. Picrotoxin (100 μM, Sigma-Aldrich) was always present in the saline to block the inhibitory inputs. Intracellular solutions were composed of (in mM): 150 Cs-gluconate, 10 HEPES, 4 MgCl$_2$, 4 Na$_2$ATP, 1 Na$_2$GTP, 0.4 EGTA and 5 lidocaine N-ethyl bromide (QX-314) (pH 7.25, 292 mOsm kg$^{-1}$). The patch pipette resistance was 2 − 4 MΩ when filled with each intracellular solution.

To evoke CF- and PF-EPSCs, square pulses (10 μs, 0 − 200 μA) were applied through a stimulating electrode placed on the granular layerand the molecular layer (∼50 μm away from the pial surface), respectively. Selective stimulation of CFs and PFs was confirmed by the paired-pulse depression (PPD) and paired-pulse facilitation (PPF) of EPSC amplitudes at a 50-ms interstimulus interval, respectively. In miniature EPSC (mEPSC) experiments, tetrodotoxin (1 μM; Alamone Lab) was applied to the extracellular solution during recordings. mEPSC traces were analyzed by using a MINI ANALYSIS program (Synaptosoft, Decatur, GA), and observed events <10 pA were discarded. The current responses were recorded using an Axopatch 200B amplifier (Molecular Devices), and the pCLAMP system (version 9.2; Molecular Devices) was used for data acquisition and analysis. The signals were filtered at 1 kHz and digitized at 4 kHz.

**FRAP analyses of labelled AMPARs in cultured neurons.** Cultured hippocampal neurons were washed with ACSF solution (120 mM NaCl, 5 mM KCl, 2 mM CaCl$_2$, 1.2 mM MgCl$_2$, 30 mM D-glucose, 25 mM HEPES and 1 μM tetrodotoxin, pH 7.4)

and treated with 1 μM **CAM2(Fl)** in ACSF solution at 17 °C for 1 h. After labelling procedure, the cells were placed on the stage of confocal microscopy (LSM710, Axio Observer.Z1, ZEISS) equipped with Definite Focus (ZEISS) (for compensating focus drifts) and continually perfused with ACSF solution. Continuous perfusion is required for removal of excess **CAM2** reagents and the resultant ligand moiety. Fluorescence was excited using 63 × oil objective (NA = 1.40) by a 488 nm line of an argon laser. Time series were collected as repetitively scanned single confocal slices. After collecting the first image, laser power was increased to 100% and a predefined circular region of interest was bleached by a single laser scan. The following image was collected within 1 s of the end of photo bleaching. Fluorescence was quantified using ZEN analysis program (ZEISS). The data were normalized to fluorescence before bleaching (defined as 1) and immediately after bleaching (0). Each FRAP plot was fitted to a single-exponential curve according to the following equation.

$$F(t) = a(1 - \exp(-bt))$$

Diffusion coefficients were determined as described previously[48].

For FRAP experiments using SEP-AMPARs, hippocampal neurons were transfected with a plasmid encoding flop form of SEP-GluA2$^{flop}$(Q) (kindly gifted from Professor Malinow) using the lipofectamine 2000 (Invitrogen) at 14 DIV and subjected to imaging experiments at 16 DIV.

For the dual colour FRAP experiment, hippocampal neurons expressing SEP-GluA2$^{flop}$(Q) were labelled with 1 μM of **CAM2(Ax647)**. After labelling procedure, the cells were placed on the stage of confocal microscopy. Fluorescence images were acquired using a 488 nm line of an argon laser for excitation of SEP and a HeNe laser for excitation of **Ax647**. After collecting the first images, the power of both laser was increased to 100% and a predefined circular region of interest was bleached by a single laser scan. The following imaging and analysis step were performed as described above.

**FRAP analyses of labelled AMPARs in hippocampal slices.** After labelling procedure, the slices were placed on the stage of upright confocal microscopy (LSM710 NLO, Axio Examiner. Z1, ZEISS) equipped with a 20 ×, numerical aperture (NA) = 1.0 water objective in ACSF solution (125 mM NaCl, 2.5 mM KCl, 2 mM CaCl$_2$, 1 mM MgCl$_2$, 1.25 mM NaH$_2$PO$_4$, 26 mM NaHCO$_3$, 10 mM D-glucose and 100 μM Picrotoxin), which was continuously bubbled with a mixture of 95% O$_2$ and 5% CO$_2$. The following imaging processes were similarly performed as described in 'FRAP analyses of labelled AMPARs in cultured neurons'.

**Data availability.** The data that support the findings of this study are available from the corresponding authors on reasonable request.

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

## Acknowledgements

We thank Roberto Malinow (University of California, San Diego) for SEP-GluA2 construct, and Eric Gouaux (Oregon Health and Science University) for the S1S2J construct. We also thank K. Akiyoshi and S. Mukai (Kyoto University) for two-photon microscopy, Y. Sato (Carl Zeiss Microscopy Inc.) for confocal microscopy, Y. Mori and S. Sawamura (Kyoto University) for neuronal culture, G. Craven (Imperial College London) for organic synthesis, E. Kusaka (Kyoto University) for NMR measurement, and M. Tsujikawa (Kyoto University) for plasmid construction. This work was funded by a Research Fellowship from the Japan Society for the Promotion of Science (JSPS) for Young Scientists (to S.W. 26-3123)), SUNBOR Grant from Suntory Foundation for Life Sciences (to S.K.), the Takeda Science Foundation (to S.K., W.K. and M.Y.), the Strategic Research Program for Brain Sciences from Japan Agency for Medical Research and Development (AMED) (to W.K.), and the Japan Science and Technology Agency (JST) Core Research for Evolutional Science and Technology (CREST) of Molecular Technologies (to I.H. and M.Y.).

## Author contributions

S.K. and I.H. initiated and designed the project. S.W. and S.K. performed synthesis and chemical labelling in test tubes or cultured cells. S.W., S.K., K.I., Y.L.N. and A.K. performed chemical labelling and live imaging in cultured neurons. I.A., W.K. and M.Y. performed electrophysiological measurements. S.W., W.K., S.M, K.I. and M.Y. performed brain slice experiments. S.K., M.Y. and I.H. wrote the manuscript. All authors discussed and commented on the manuscript.

## Additional information

**Competing interests:** The authors declare no competing financial interests.

