## [Peer Review File · Nature Communications]

Reviewers' Comments:

Reviewer #1 (Remarks to the Author)

In this manuscript, Wakayama et al. developed an approach (CAM) to chemically label endogenous AMPA receptors with small fluorophores in both cell cultures and brain slices. They confirmed that the labeling is specific and the receptor function is not affected by the labeling. The authors used this method to characterize the diffusion dynamics of AMPARs and report different results from that found using SEP-AMPA. This is an impressive technique that could have a significant impact on the field especially if it can be translated to other receptors.

A way to label endogenous AMPAR with small fluorophores is appealing to many researchers in the field. The approach presented in this study is promising. However, there remain a few concerns and questions as follows.

1. One concern is what population of AMPARs is labeled. Are only surface AMPARs labeled? What percentage of AMPARs is labeled? Are different AMPAR subunits (GluA1-4) labeled equally? The answers to these questions are very important to explain the results obtained with this method.
2. The labeling was not performed under physiological conditions. For labeling of hippocampal neurons, the cells were stained in Neurobasal medium at 17 C for 1-4 h. The irregular appearance of MAP2 staining along dendrites indicates very unhealthy neurons. The authors should perform staining under more physiological conditions. And they should also assess whether the staining is specific and surface-only under these conditions.
3. The authors used immunostaining to show that the CAM ligand specifically labels surface AMPARs (Fig. 3 and 4). However, the staining for GluA2/3 was performed after permeabilization. It's better to perform surface immunostaining for AMPARs for this comparison.
4. For FRAP experiments, the CAM stained and SEP-AMPA neurons were subjected to different experimental procedures. For CAM labeling, the neurons were placed in 17 C for 1 h while SEP-AMPA transfected neurons were not. Could this be the reason for the observed difference in diffusion dynamics? The author should perform experiments for both groups under the same conditions. A very nice experiment to do will be a two-color experiment: Label SEP-AMPA transfected neurons with a CAM-Ax647 and image diffusion dynamics in both color channels after photobleaching.
5. In addition it is not clear how the FRAP experiments were performed. Was the SEP-GluA2 transfected? For how long? Transiently transfected receptors may behave differently than endogenous receptor despite there the SEP tag due to the time of transfection and the formation of nonphysiological tetramers (GluA2 homomers). The CAM stained receptors may have different subunit combinations and thus behave differently in the FRAP studies. Thus, the conclusions that there are differences between SEP-tagged and CAM labeled receptors should be toned down.
6. Most of the WBs show very clean background. Were the contrast adjusted properly?
7. Fig. 4c: the Colocalization between Fl and anti-GluA2/3 is not clear. They should show higher magnification images.
8. Fig. 5d: the authors should show the average cumulative distribution of amplitude for control and labeled group. And they should do Kolmogorov-Smirnov test for statistical significance instead of only comparing the average amplitudes.

9. In the introduction of the manuscript, the authors say that some conflicting findings for AMPAR trafficking are due to limitations of available methods. However, in the end, they did not show how the newly developed method could solve the conflicts.

Reviewer #2 (Remarks to the Author)

In this submitted manuscript, "Chemical labeling for visualizing native AMPA receptors in live neurons", Sho Wakayama et.al presented a traceless protein labeling method 'LDAI', for labeling and visualizing native significant receptor AMPAR. Based on the LDAI strategy, they designed "CAM" reagents consisting of AMPAR ligand, reactive site (Acyl imidazole), spacer and probes for various purposes. Then they successfully applied this tool for labeling and visualizing AMPAR expressed on HEK293T cells, native AMPAR in cultured neurons as well as in brain tissues. At the aspect of labeling mechanism, the authors validated the 'nucleophilic attack site' in the LDAI labeling process and listed several advantages for this CAM reagents. At the physiological function aspect, they have provided several experiments to demonstrate that this labeling method has negligible effects on AMPAR's function. Last but not least, they combined this labeling strategy with FRAP method for diffusion dynamics study of AMPAR which allowed them to declare that it is the first report showing constitutive diffusion dynamics of endogenous AMPARs in three-dimensional brain slices. Altogether, the authors demonstrated the advantages and significance of this chemical labeling method for visualizing AMPAR by providing convincing results, which makes this manuscript a good candidate for publication.

With that being said, there are still some concerns need to be addressed. Both the LDAI method and the study object AMPAR appeared in the previous papers from the same group (J. Am. Chem. Soc. 134, 3961–3964 (2012)(Ref.11); Chem. Biol. 21, 1013–1022 (2014)(Ref.12); DOI: 10.1038/NCHEM.2554). In particular, the LDAI method have already been applied on several membrane receptors and some of its advantages have been indicated in these previous work. Despite the 'LDAI' reagent was designed for another protein AMPAR, this labeling method can hardly be named as a 'new' tool in this manuscript. While we still think the following labeling results in neurons and tissues successfully demonstrated the power of this labeling tool.

More specific questions:

1. The authors studied the diffusion dynamics of AMPAR both in cultured neurons and brain slices. As mentioned in their discussion session, can the authors demonstrate additional endocytosis or recycling assays of native AMPARs in cultured neurons?
2. Can the CAM reagents bypass the blood-brain barrier or decompose in circulation? The author may inject the reagent into the tail vein of mice and detect the fluorescence of AMPAR in brain tissues to address this issue.
3. The fluorescent dots in Fig 4b is totally different from those in Fig 4c while the schematic figures showed the same image area, which is confusing. Despite the different experimental conditions between these two assays, I guess the differences resulted from the amplification factors. Thus we suggested that the two schematic figures should be modified as the schematic figures shown in Supplementary Figure 14.
4. The authors declared that CAM2 reagent has no effect on synaptic function in Figure 5. While the traces of control and CAM2-labeled mEPSC shown in Figure 5c are not exactly the same. How to explain the difference?

Reviewer #3 (Remarks to the Author)

Wakayama et al. describe a method for optically monitoring the number and location (trafficking) of native AMPA receptors in isolated cells and brain tissue using a covalent chemical labeling strategy that tethers small fluorophores to non-functional sites on the AMPARs. The methodology and data are appropriate and of high quality, and the conclusions drawn appear valid. The paper is well-written.

AMPA receptor trafficking is the cornerstone of many forms of adaptive and maladaptive neuronal plasticity. The chemical AMPAR modification (CAM) reagents are expected to advance the field.

Minor Concerns:

1) Please clarify what construct - GluA2R or GluA2Q - was used for the HEK experiments and why this subunit was chosen. The description of the HEK experiments starts on p7 line 1 (cells were transiently transfected with the "GluA2 AMPAR subtype") but then on p8 line 21 the GluA2(Q) subtype is mentioned. Was the GluA2(Q) subtype used for ALL of the HEK cell experiments? If so, please state this at the beginning of the section for clarity. Also, GluA2 homomeric receptors are not thought to be expressed in adult animals, and are edited at the mRNA level to produce GluA2(R) subunits (which combine with GluA1, 3, and 4 subunits to form heteromeric channels in vivo). So, why was the GluA2(Q) subunit chosen to characterize the CAMs in HEK cells? This is a bit odd, so the logic should be presented and reference to previously studies of GluA2 homomers made (see Dingledine, Washburn and citing references).

2) Literature citations are generally appropriate, though review articles are sometimes used in place of the primary literature (e.g. refs 17-19). Best to also include the primary literature citations, or indicate in the text that the "conflicting findings" are reviewed in references 17-19. This is a small point, but will help the interested reader identify the "conflicts" more readily and thus appreciate the potential value of CAMs.

Response to Reviewer 1's comments

Comments:

In this manuscript, Wakayama et al. developed an approach (CAM) to chemically label endogenous AMPA receptors with small fluorophores in both cell cultures and brain slices. They confirmed that the labeling is specific and the receptor function is not affected by the labeling. The authors used this method to characterize the diffusion dynamics of AMPARs and report different results from that found using SEP-AMPA. This is an impressive technique that could have a significant impact on the field especially if it can be translated to other receptors.

A way to label endogenous AMPAR with small fluorophores is appealing to many researchers in the field. The approach presented in this study is promising. However, there remain a few concerns and questions as follows.

Our response:

We would like to thank the reviewer for kind review and for important comments. According to the suggestions and comments, we have carefully amended our manuscript as shown below. All the revisions we made are highlighted in red in the revised manuscript.

Comment 1

1) One concern is what population of AMPARs is labeled. Are only surface AMPARs labeled? What percentage of AMPARs is labeled? Are different AMPAR subunits (GluA1-4) labeled equally? The answers to these questions are very important to explain the results obtained with this method.

Comment 1-1: Are only surface AMPARs labeled?

Our response:

In our original manuscript, we had already revealed that our labeling reagents did not permeate into live cells and labeled surface-exposed AMPARs selectively in HEK cells (**Supplementary Fig. 4**). To further confirm surface AMPAR labeling, we newly conducted fluorescent quenching experiments using the vital dye trypan blue (TB) which is excluded from live cells as reported previously (refs. 43–45).

In HEK293T cells, addition of TB immediately after **CAM2**-labeling successfully quenched most of the labeled fluorescence (**Supplementary Fig. 11a**). In contrast, when TB was applied to

cells that were incubated at 37 °C to facilitate internalization of AMPARs (ref. 37), the fluorescence from the intracellular space was not quenched, whereas the fluorescence from the cell surface was mostly quenched in this condition (**Supplementary Fig.11b, c**). These results indicate that TB can be used to selectively quench the fluorescence from the surface-exposed AMPARs but not the internalized ones.

In live neurons, we found that TB-treatment immediately after **CAM2**-labeling largely abolished the labeled fluorescence. Importantly, the quenching ratio ($I_{TB}/I_{initial}$) in neurons was comparable to that obtained in HEK293T cells (**Supplementary Fig. 11d, e**). In contrast, the fluorescence quenching was significantly impaired after facilitation of internalization of labeled AMPARs by incubation at 37 °C, indicating that TB treatment selectively quenches surface-exposed receptors in neurons (**Supplementary Fig. 11f, g**). These results indicate that the surface-exposed AMPARs are mainly visualized in live neurons by our method.

Modification in the main text

In page 7, line 144: As expected, **CAM2(OG)** did not permeate into live cells (**Supplementary Fig. 4a**), and selectively labeled cell-surface GluA2 (**Supplementary Fig. 4b**).

In page 10, line 217: To examine whether the **FI** signals were derived from the cell surface AMPARs, fluorescence quenching experiments were conducted using the vital dye trypan blue⁴³⁻⁴⁵. Similar to the results of HEK293T cells in which trypan blue selectively quenched the fluorescence from the surface-exposed AMPARs but not the internalized ones (**Supplementary Fig. 11a-c**), addition of trypan blue largely abolished the fluorescence immediately after **CAM2** labeling in cultured hippocampal neurons (**Supplementary Fig. 11d-g**). These results indicate that the surface-exposed AMPARs were predominantly labeled by our methods.

Comment 1-2: What percentage of AMPARs is labeled?

Our response:

We determined that the labeling yield of GluA2 subunit as 62.0 ± 2.4 % in HEK293T cells, which had already been described in **Supplementary Figure 7** in the original manuscript. To avoid confusion, we have cited this figure (now **Supplementary Figure 2**) in the section of “live imaging”, and described the labeling yield in the legend to **Figure 2** of the revised manuscript.

With respect to labeling yield to GluA2 subunit in the live imaging condition using neuronal culture, we newly added the data in **Supplementary Figure 20** to respond to this reviewer’s

comment, and described the labeling yield in **Figure 6** legend.

Modification in the main text

In page 7, line 140: Live-cell confocal imaging of GluA2-expressing HEK293T cells labeled with **CAM2(OG)** clearly showed that the fluorescence was observed predominantly at the plasma membrane (**Fig. 2c** and **Supplementary Fig. 2**),

In page 42, line 949: Labeling yield of **CAM2** to surface-exposed GluA2 in this condition was determined as 62.0 ± 2.4 % (**Supplementary Fig. 2**).

In page 46, line 1037: Labeling yield of **CAM2** to surface-exposed GluA2 in this condition was determined as 9.6 ± 0.9 % (**Supplementary Fig. 20**).

Comment 1-3: Are different AMPAR subunits (GluA1-4) labeled equally?

Our response:

According to this reviewer's comment, we examined chemical labeling of each AMPAR subunit (GluA1–4) expressed in HEK293T cells, and determined the relative labeling efficiency by western blotting (**Supplementary Fig. 7**). The results indicated that **CAM2** reagent successfully labeled GluA2, GluA3 and GluA4 but not GluA1, although GluA1 was prominently expressed on the cell surface like GluA2. The phylogenetic tree reveals low homology of GluA1 relative to the other AMPAR subunits, and we found that two of the three labeled sites identified in GluA2 are not conserved in GluA1. Thus, it may be considered that differences in the microenvironment of the entrance of the ligand-binding pocket inhibit the chemical labeling to GluA1 using **CAM2**.

Modification in the main text

In page 8, line 163: A functional AMPAR is a tetramer consisting of a combination of four subunits, GluA1–GluA4^{23,24,39}. To examine selectivity of **CAM2** reagents to GluA subunits, HEK293T cells were transfected with each subunit (GluA1, GluA2, GluA3 or GluA4). WB analysis revealed that **CAM2** successfully labeled GluA2, GluA3 and GluA4 but not GluA1 (**Supplementary Fig. 7a–d**), although GluA1 was prominently expressed on the cell surface like GluA2 (**Supplementary Fig. 7e**). The phylogenetic tree indicates the low homology of GluA1 among the AMPAR subunits (GluA1–4) (**Supplementary Fig. 7f**), and two of the three labeling sites identified for GluA2 are not conserved in GluA1 (**Fig. 2b** and **Supplementary Fig. 7g**). Such difference in the microenvironment of the entrance of the ligand-binding pocket may inhibit the GluA1 labeling by

CAM2.

In page 9, line 190: Together, these data indicate that **GluA2, GluA3 and GluA4 subunits of AMPARs** on the plasma membrane can be selectively labeled near the ligand-binding domain with a small fluorophore using **CAM** reagents under live-cell conditions with negligible disturbance of receptor function.

Comment 2

2) *The labeling was not performed under physiological conditions. For labeling of hippocampal neurons, the cells were stained in Neurobasal medium at 17 C for 1-4 h. The irregular appearance of MAP2 staining along dendrites indicates very unhealthy neurons. The authors should perform staining under more physiological conditions. And they should also assess whether the staining is specific and surface-only under these conditions.*

Our response:

To address the reviewer's concern, we carefully re-examined the immunocytochemical staining patterns of neurons labeled by CAM2. We found that under optimal conditions, MAP2-immunopositive neurons appeared normal. In addition, the labeled signals merged well with punctate anti-PSD95 signals, and broadly colocalized with anti-GluA2 signals. Moreover, the surface immunostaining of anti-GluA2 merged well with the **CAM2**-labeled signals, which is described in details in our response to your comment #3. To avoid confusion, we have replaced previous images with the more representative ones in the revised manuscript (**Fig. 3d-h**).

With respect to a concern about influences on neuronal function by our labeling procedure, we already showed electrophysiological properties of Purkinje cells labeled with the same condition in the original manuscript (**Fig. 5**), indicating that the **CAM2** labeling procedure unlikely affected neuronal functions. Therefore, we did not modify the labeling conditions (*at 17 °C for 1-4 h*) under which internalization of AMPARs can be suppressed (ref. 37).

Modification in the main text

In page 11, line 224: To characterize the **FI** signals **in details**, we next performed conventional immunohistochemical analyses on hippocampal neurons labeled by **CAM2(FI)** after fixation and permeabilization. Confocal microscopy images showed punctate **FI** signals located on protrusions

along dendrites immunopositive for microtubule-associated protein 2 (MAP2) (**Fig. 3d** and **3f**). **The FI signals merged well with punctate immunopositive signals for postsynaptic density protein 95 (PSD95; Fig. 3h), and also broadly colocalized with immunoreactivity for GluA2/3 (Fig. 3e and 3g).**

Comment 3

3. *The authors used immunostaining to show that the CAM ligand specifically labels surface AMPARs (Fig. 3 and 4). However, the staining for GluA2/3 was performed after permeabilization. It's better to perform surface immunostaining for AMPARs for this comparison.*

Our response:

According to the reviewer's advice, we examined surface immunostaining of AMPARs in cultured hippocampal neurons labeled with **CAM2** without the permeabilization treatment. As shown in **Supplementary Figure 12**, the immunostained punctate signals of surface AMPARs co-localized well with the CAM2-labeled (**FI**) signals.

Modification in the main text

In page 11, line 230: **Importantly, immunostained punctate signals of surface AMPARs co-localized well with the FI signals (Supplementary Fig. 12).**

Comment 4

4. *For FRAP experiments, the CAM stained and SEP-AMPA neurons were subjected to different experimental procedures. For CAM labeling, the neurons were placed in 17 C for 1 h while SEP-AMPA transfected neurons were not. Could this be the reason for the observed difference in diffusion dynamics? The author should perform experiments for both groups under the same conditions. A very nice experiment to do will be a two-color experiment: Label SEP-AMPA transfected neurons with a CAM-Ax647 and image diffusion dynamics in both color channels after photobleaching.*

Our response:

We appreciate this valuable advice. According to the reviewer's suggestion, we newly conducted

two-color FRAP experiments. Hippocampal neurons exogenously expressed with SEP-AMPA were labeled with CAM2(Ax647), and two-color FRAP analysis was performed in the same dendritic spines. As shown in **Supplementary Figure 21**, the diffusion dynamics of Ax647-AMPA in neurons expressing SEP-AMPA were comparable to that of FI-AMPA obtained in non-transfected neurons, indicating the substantially smaller recovery ratio relative to that obtained by SEP-AMPA. This result indicates that the high recovery ratio and slow kinetics of the SEP-AMPA could not be ascribed to overexpression of AMPA, supporting our view that other trafficking processes are involved owing to the pH-sensitivity of SEP in the analysis of SEP-AMPA.

Modification in the main text

In page 14, line 317: To test the possibility that such differences were caused by overexpression of AMPA in the analysis of SEP-AMPA, we next labeled hippocampal neurons expressing SEP-AMPA with CAM2(Ax647). The dual color FRAP analysis revealed that the diffusion dynamics of Ax647-AMPA in neurons expressing SEP-AMPA were comparable to that of FI-AMPA obtained in non-transfected neurons (**Supplementary Figure 21**). These results indicate that high recovery ratio and slow kinetics of the SEP-AMPA could not be ascribed to overexpression of AMPA. A plausible explanation is the involvement of other trafficking processes, such as those from intracellular organelle to the surface owing to the pH-sensitivity of SEP (for details, see Discussion).

Comment 5

5. In addition it is not clear how the FRAP experiments were performed. Was the SEP-GluA2 transfected? For how long? Transiently transfected receptors may behave differently than endogenous receptor despite there the SEP tag due to the time of transfection and the formation of nonphysiological tetramers (GluA2 homomers). The CAM stained receptors may have different subunit combinations and thus behave differently in the FRAP studies. Thus, the conclusions that there are differences between SEP-tagged and CAM labeled receptors should be toned down.

Our response:

According to this reviewer's comment, we have added detailed description of the FRAP experiments in the Method section "FRAP analyses of labeled AMPA in cultured neurons" of

the revised manuscript. In addition, we have toned down the conclusion about the differences between SEP-tagged and CAM-labeled receptors in Discussion section as suggested by the reviewer.

Modification in the main text

In page 18, line 393: **Moreover, exogenously expressed AMPARs may behave differently than endogenous AMPARs due to the formation of nonphysiological tetramers. Thus, careful consideration would be needed to assess** the diffusion dynamics of SEP-AMPARs.

In page 31, line 720: For FRAP experiments using SEP-AMPARs, hippocampal neurons were transfected with a plasmid encoding flop form of SEP-GluA2^{flop(Q)} (kindly gifted from Prof. Malinow) using the lipofectamine 2000 (Invitrogen) **at 14 DIV** and subjected to imaging experiments **at 16 DIV**.

For the dual color FRAP experiment, hippocampal neurons expressing SEP-GluA2^{flop(Q)} were labeled with 1 μM of CAM2(Ax647). After labeling procedure, the cells were placed on the stage of confocal microscopy. Fluorescence images were acquired using a 488 nm line of an argon laser for excitation of SEP and a HeNe laser for excitation of Ax647. After collecting the first images, the power of both laser was increased to 100% and a predefined circular region of interest was bleached by a single laser scan. The following imaging and analysis step were performed as described above.

Comment 6

6. *Most of the WBs show very clean background. Were the contrast adjusted properly?*

Our response:

In our WB analysis, the original data were analyzed using analytical software 'ImageQuant FL' which is attached to a CCD imager 'ImageQuant LAS 4000' (GE healthcare). In the analysis, we carefully and correctly adjusted the contrast. For your information, the analytical process of one of the representative data (**Figure 4a CAM2(FI)**) is shown as below.

Comment 7

7. Fig. 4c: the Colocalization between Fl and anti-GluA2/3 is not clear. They should show higher magnification images.

Our response:

According to this reviewer's comment, we have changed the original **Figure 4c** into the magnified one in the revised manuscript.

Comment 8

8. Fig. 5d: the authors should show the average cumulative distribution of amplitude for control and labeled group. And they should do Kolmogorov–Smirnov test for statistical significance instead of only comparing the average amplitudes.

Our response:

According to this reviewer's comment, we performed Kolmogorov–Smirnov test in **Figure 5d** data, and significant differences were not observed with or without the chemical labeling. This supports that miniature EPSCs are unaffected by **CAM2** labeling.

Modification in the main text

In page 46, line 1029: Kolmogorov–Smirnov test in **d** and Mann–Whitney *U* test in **e** indicate that significant differences were not observed with or without the chemical labeling.

Comment 9

9. *In the introduction of the manuscript, the authors say that some conflicting findings for AMPAR trafficking are due to limitations of available methods. However, in the end, they did not show how the newly developed method could solve the conflicts.*

Our response:

According to this reviewer’s comment, we have added a description about utilization of our labeling method to solve the conflicting AMPAR trafficking in “Discussion section”.

Modification in the main text

In page 18, line 396: Trafficking of AMPARs is dynamically regulated during synaptic plasticity, and the number of postsynaptic AMPARs is tightly regulated by the balance between insertion and internalization of receptors. To visualize these events in live neurons, SEP-tagged AMPARs have been widely utilized^{3,4}. However, it is recently pointed out that pH changes in the intracellular pools could affect SEP-AMPA fluorescent signals³³. Since **CAM2** predominantly labeled surface-exposed AMPARs, it is expected to serve as a simple tool to monitor the trafficking of cell surface AMPARs in live neurons.

Response to Reviewer 2's comments

Comments:

In this submitted manuscript, “Chemical labeling for visualizing native AMPA receptors in live neurons”, Sho Wakayama et.al presented a traceless protein labeling method ‘LDAI’, for labeling and visualizing native significant receptor AMPAR. Based on the LDAI strategy, they designed “CAM” reagents consisting of AMPAR ligand, reactive site (Acyl imidazole), spacer and probes for various purposes. Then they successfully applied this tool for labeling and visualizing AMPAR expressed on HEK293T cells, native AMPAR in cultured neurons as well as in brain tissues. At the aspect of labeling mechanism, the authors validated the ‘nucleophilic attack site’ in the LDAI labeling process and listed several advantages for this CAM reagents. At the physiological function aspect, they have provided several experiments to demonstrate that this labeling method has negligible effects on AMPAR’s function. Last but not least, they combined this labeling strategy with FRAP method for

diffusion dynamics study of AMPAR which allowed them to declare that it is the first report showing constitutive diffusion dynamics of endogenous AMPARs in three-dimensional brain slices. Altogether, the authors demonstrated the advantages and significance of this chemical labeling method for visualizing AMPAR by providing convincing results, which makes this manuscript a good candidate for publication.

Our response:

We appreciate kind review and important comments by this reviewer. We have amended our manuscript as shown below. All the revisions we made are highlighted in red in the revised manuscript.

General Comment

With that being said, there are still some concerns need to be addressed. Both the LADI method and the study object AMPAR appeared in the previous papers from the same group (J. Am. Chem. Soc. 134, 3961–3964 (2012)(Ref.11); Chem. Biol. 21, 1013–1022 (2014)(Ref.12); DOI: 10.1038/NCHEM.2554). In particular, the LADI method have already been applied on several membrane receptors and some of its advantages have been indicated in these previous work. Despite the ‘LADI’ reagent was designed for another protein AMPAR, this labeling method can

hardly be named as a 'new' tool in this manuscript. While we still think the following labeling results in neurons and tissues successfully demonstrated the power of this labeling tool.

Our response:

As commented by this reviewer, we have previously reported membrane protein labeling using LDAI methods. However, in our previous papers, we have not succeeded in the selective protein labeling and imaging of endogenous receptors in neurons and more complicated tissues. Therefore, this is the first report for selective chemical labeling and visualization of endogenous neurotransmitter receptors with active forms in live neurons and live neuronal tissues. Nevertheless, according to this reviewer's comment, we deleted the word "new" or "novel" in Abstract, Introduction and Discussion part in this revised manuscript.

Modification in the main text

In page 2, line 38: **This method will help clarify the involvement of AMPAR trafficking in various neuropsychiatric and neurodevelopmental disorders.**

In page 5, line 93: In the present study, we report a **promising** traceless protein labeling method that effectively tethers various small fluorescent probes to endogenous AMPARs located at the cell surface without affecting AMPAR function.

In page 16, line 339: In this report, we described the development of **useful** chemical tools for the selective labeling and imaging of endogenous AMPARs in live cultured neurons and brain tissues.

More specific comment 1

1. The authors studied the diffusion dynamics of AMPAR both in cultured neurons and brain slices. As mentioned in their discussion session, can the authors demonstrate additional endocytosis or recycling assays of native AMPARs in cultured neurons?

Our response:

According to the suggestion by this reviewer, we have newly conducted endocytosis assays of native AMPARs in cultured neurons and added these results in the main text. Here, we focused on visualization of trafficking of **CAM2**-labeled AMPAR during long-term depression (LTD), a well-known synaptic plasticity. It is reported that synaptic AMPARs are internalized to decrease the surface exposed AMPARs during LTD. To visualize this pathway, native AMPARs were labeled

with fluorescein, a pH-sensitive dye. As shown in **Supplementary Figure 14**, the prominent fluorescent decrease was observed in the chemically induced form of LTD (chemLTD). Taking into consideration of the pH sensitivity of **FI**-labeled AMPARs on cell surface (**Supplementary Fig. 15**), the fluorescence decrease indicates that we succeeded in imaging of endocytosis of native AMPARs in live neurons using our methods.

Modification in the main text

In page 11, line 239: We next sought to follow the trafficking of **FI**-labeled AMPARs during long-term depression (LTD), a well-known synaptic plasticity^{23,24}. To apply chemically induced form of LTD (chemLTD)⁴⁶, the labeled hippocampal neurons were exposed with NMDA in a short period, and fluorescent changes of **FI**-labeled AMPARs were visualized by confocal live imaging. As shown in **Supplementary Figure 14**, the fluorescence decrease was observed in punctate regions after brief application of NMDA. Taking into consideration of the pH sensitivity of **FI**-labeled AMPARs on cell surface (**Supplementary Fig. 15**), the fluorescent change implies the internalization of AMPARs into acidic intracellular endosomes after chemLTD, which is in good agreement with previous reports³.

More specific comment 2

2. *Can the CAM reagents bypass the blood-brain barrier or decompose in circulation? The author may inject the reagent into the tail vein of mice and detect the fluorescence of AMPAR in brain tissues to address this issue.*

Our response:

We appreciate this critical comment. Actually this is one of our final goals in this project. However, to examine *in vivo* labeling, we still need huge additional experiments (e.g. stability of our reagents in blood, ability to bypass the blood-brain barrier, and toxicity for live mice, etc). Apparently, these are different scopes compared with the main claims in this paper. We will perform these experiments and would like to report it in the future.

More specific comment 3

3. *The fluorescent dots in Fig 4b is totally different from those in Fig 4c while the schematic*

figures showed the same image area, which is confusing. Despite the different experimental conditions between these two assays, I guess the differences resulted from the amplification factors. Thus we suggested that the two schematic figures should be modified as the schematic figures shown in Supplementary Figure 14.

Our response:

We thank the reviewer for this comment. We are sorry that the schematic illustration in **previous Figure 4c** was wrong. According to the reviewer's comment, schematic illustrations in **Figure 4b** and **4c** have been modified as shown in **Supplementary Figure 19** (corresponds to previous **Supplementary Figure 14**) in the revised manuscript.

More specific comment 4

4. The authors declared that CAM2 reagent has no effect on synaptic function in Figure 5. While the traces of control and CAM2-labeled mEPSC shown in Figure 5c are not exactly the same. How to explain the difference?

Our response:

It is known that the trace shape of miniature EPSC does not overlap in each experiment even in the same neurons. Thus, the average amplitude and average cumulative distribution of amplitude are statistically compared in general. As shown in Figure **5d** and **5e**, any significant differences in these analyses were not observed with or without the chemical labeling. This strongly supports that miniature EPSCs are unaffected by **CAM2** labeling.

Modification in the main text

In page 46, line 1029: **Kolmogorov–Smirnov test in d and Mann-Whitney U test in e indicate that significant differences were not observed with or without the chemical labeling.**

Response to Reviewer 3's comments

Comments:

Wakayama et al. describe a method for optically monitoring the number and location (trafficking) of native AMPA receptors in isolated cells and brain tissue using a covalent chemical labeling strategy that tethers small fluorophores to non-functional sites on the AMPARs. The methodology and data are appropriate and of high quality, and the conclusions drawn appear valid. The paper is well-written.

AMPA receptor trafficking is the cornerstone of many forms of adaptive and maladaptive neuronal plasticity. The chemical AMPAR modification (CAM) reagents are expected to advance the field.

Our response:

We thank reviewer #3 for kind review and for important comments. We have carefully taken those comments into consideration and amended our manuscript as shown below. All the revisions we made are highlighted in red in the revised manuscript.

Minor Comment-1

1) Please clarify what construct - GluA2R or GluA2Q - was used for the HEK experiments and why this subunit was chosen. The description of the HEK experiments starts on p7 line 1 (cells were transiently transfected with the "GluA2 AMPAR subtype") but then on p8 line 21 the GluA2(Q) subtype is mentioned. Was the GluA2(Q) subtype used for ALL of the HEK cell experiments? If so, please state this at the beginning of the section for clarity. Also, GluA2 homomeric receptors are not thought to be expressed in adult animals, and are edited at the mRNA level to produce GluA2(R) subunits (which combine with GluA1, 3, and 4 subunits to form heteromeric channels in vivo). So, why was the GluA2(Q) subunit chosen to characterize the CAMs in HEK cells? This is a bit odd, so the logic should be presented and reference to previously studies of GluA2 homomers made (see Dingledine, Washburn and citing references).

Our response:

In most of HEK experiments, GluA2(R), a representative subtype of AMPARs was used. However, in Ca²⁺ imaging and electrophysiological assays, we used GluA2(Q) because AMPARs containing GluA2(R) did not show Ca²⁺ permeability. To avoid confusion, we have added explanation about

GluA2(Q) at the beginning of the Ca²⁺ imaging section, and we have added description of GluA2(R) or GluA2(Q) in the all figure legend. In addition, we have also described flop or flip splice variants in Methods or Figure legends.

As suggested by this reviewer, we have cited references about previous studies of GluA2 homomers.

Modification in the main text

In page 7, line 126: Covalent labeling of AMPARs was initially examined in live HEK293T cells transiently transfected with the **GluA2 (GluA2(R)) subtype, a major subtype of AMPARs in brains.**

In page 9, line 178: To address this concern, we first performed Ca²⁺ imaging and found that **CAM2(OG)** did not affect Ca²⁺ responses in HEK293T cells expressing Ca²⁺-permeable AMPARs (GluA2(Q))⁴⁰⁻⁴² (**Supplementary Fig. 8**).

In page 19, line 422: Utilizing a PCR method, cDNA encoding an HA tag was added to the 5' end (immediately following the signal sequence) of mouse GluA2^{flop}(R), GluA2^{flip}(Q), or GluA2^{flop}(Q), **GluA1^{flip}(Q), GluA3^{flip}(Q) or GluA4^{flip}(Q)** cDNA.

In page 42, line 935: HEK293T cells transfected with GluA2^{flop}(R) (AMPA(+)) or the control vector (AMPA(-)) were treated with 2 μM labeling reagents **CAM1(OG), CAM2(OG), or CAM3(OG)** in the presence or absence of 50 μM NBQX in serum free DMEM.

In page 43, line 952: Current responses to 80-ms applications of glutamate using piezo element (during upper step pulses) were obtained in outside-out patches from HEK293T cells expressing GluA2^{flop}(Q) with or without labeling procedures using **CAM2(Ax488)** (For details, see Methods).

Minor Comment-2

2) *Literature citations are generally appropriate, though review articles are sometimes used in place of the primary literature (e.g. refs 17-19). Best to also include the primary literature citations, or indicate in the text that the "conflicting findings" are reviewed in references 17-19. This is a small point, but will help the interested reader identify the "conflicts" more readily and thus appreciate the potential value of CAMs.*

Our response:

According to the reviewer's comment, we have cited primary literature in addition to review articles

in the revised manuscript.

Modification in the main text

In page 3, line 62: To reduce the size of the protein tags (20–33 kDa), a complementary recognition pairs comprising a short peptide tag (1–3 kDa) and a small molecular probe are also being developed⁹⁻¹³.

In page 4, line 89: Nevertheless, conflicting findings for AMPAR trafficking have been reported²⁵⁻³³, most likely reflecting the limitations of currently available methods.

REVIEWERS' COMMENTS:

Reviewer #1 (Remarks to the Author):

The revised version of the manuscript has addressed many of my original comments and is much improved.

Reviewer #2 (Remarks to the Author):

The authors have addressed all of previous concerns by supplementing solid experimental results and careful context modifications. More supporting experiments have been demonstrated to consolidate those appropriate results the author proposed in the revised manuscript. Some further applications have also been done, such as endocytosis assays. The revised manuscript has been carefully examined not only in experiment description but also in conclusions. Accordingly, I think this new revision is suitable for publication in Nature Communications.

Reviewer #3 (Remarks to the Author):

Wakayama et al. describe a method for optically monitoring the number and location (trafficking) of native AMPA receptors in isolated cells and brain tissue using a covalent chemical labeling strategy that tethers small fluorophores to non-functional sites on the AMPARs.

The authors have adequately addressed reviewer concerns. The manuscript is now suitable for publication.